# Effect of 17β-estradiol on the daily pattern of ACE2, ADAM17, TMPRSS2 and estradiol receptor transcription in the lungs and colon of male rats

**Iveta Herichová** ⓘ *, **Soňa Jendrisková, Paulína Pidíková, Lucia Kršková, Lucia Olexová, Martina Morová, Katarína Stebelová, Peter Štefánik**

Department of Animal Physiology and Ethology, Faculty of Natural Sciences, Comenius University, Bratislava, Slovak Republic

* herichova1@uniba.sk

**Data Availability Statement:** All relevant data are within the paper and its Supporting Information files.

## Abstract

Covid-19 progression shows sex-dependent features. It is hypothesized that a better Covid-19 survival rate in females can be attributed to the presence of higher 17β-estradiol (E2) levels in women than in men. Virus SARS-CoV-2 is enabled to enter the cell with the use of angiotensin converting enzyme 2 (ACE2). The expression of several renin-angiotensin system components has been shown to exert a rhythmic pattern, and a role of the circadian system in their regulation has been implicated. Therefore, the aim of the study is to elucidate possible interference between E2 signalling and the circadian system in the regulation of the expression of ACE2 mRNA and functionally related molecules. E2 was administered at a dosage of 40 µg/kg/day for 7 days to male Wistar rats, and sampling of the lungs and colon was performed during a 24-h cycle. The daily pattern of expression of molecules facilitating SARS-CoV-2 entry into the cell, clock genes and E2 receptors was analysed. As a consequence of E2 administration, a rhythm in ACE2 and TMPRSS2 mRNA expression was observed in the lungs but not in the colon. ADAM17 mRNA expression showed a pronounced rhythmic pattern in both tissues that was not influenced by E2 treatment. ESR1 mRNA expression exerted a rhythmic pattern, which was diminished by E2 treatment. The influence of E2 administration on ESR2 and GPER1 mRNA expression was greater in the lungs than in the colon as a significant rhythm in ESR2 and GPER1 mRNA expression appeared only in the lungs after E2 treatment. E2 administration also increased the amplitude of *bmal1* expression in the lungs, which implicates altered functioning of peripheral oscillators in response to E2 treatment. The daily pattern of components of the SARS-CoV-2 entrance pathway and their responsiveness to E2 should be considered in the timing of pharmacological therapy for Covid-19.

## Introduction

Severe acute respiratory syndrome coronavirus-2 (SARS-CoV-2), the causative agent of coronavirus disease 2019 (Covid-19), enters the cell using angiotensin converting enzyme 2

**Funding:** This work was supported by APVV-16-0209, The Slovak Research and Development Agency, https://www.apvv.sk/?lang=en, to IH; APVV-20-0241, The Slovak Research and Development Agency, https://www.apvv.sk/?lang=en, to IH; VEGA 1/0679/19, Scientific Grant Agency of the Ministry of Education, Science, Research and Sport of the Slovak Republic, https://www.minedu.sk/vedecka-grantova-agentura-msvvas-sra-sav-vega/, to IH; Operation Program of Integrated Infrastructure for the project, Advancing University Capacity and Competence in Research, Development and Innovation, ITMS2014+: 313021X329, co-financed by the European Regional Development Fund to LO. The funders had no role in study design, data collection and analysis, decision to publish, or preparation of the manuscript.

**Competing interests:** The authors have declared that no competing interests exist.

(ACE2). This requires activation of spike protein present in the envelope of the virion, which is usually conducted by transmembrane protease/serine protease 2 (TMPRSS2). Another molecule that has a strong impact on SARS-CoV-2 entry into the cell is ADAM metallopeptidase domain 17 (ADAM17) that cleaves catalytically active ectodomain of ACE2 from the cell membrane, which prevents binding and fusion of SARS-CoV-2 with the cell [1, 2]. Although the importance of ACE2-mediated SARS-CoV-2 cell entry is obvious [3], alternative ways of SARS-CoV-2 penetration have been suggested [4, 5].

In spite of the above-described role of ACE2 as a receptor in SARS-CoV-2 cell invasion, downregulation of ACE2 in the respiratory tract is not beneficial for patients [6]. ACE2 contributes significantly to blood pressure regulation, mediated by pro-inflammatory vasoconstrictor angiotensin II (AngII) by its conversion into vasodilatator angiotensin(1–7) [1, 7]. Therefore, manipulating of ACE2 and ADAM17 levels is recently considered as a new approach in Covid-19 treatment [6, 8].

In addition to the most often observed Covid-19 symptoms like lungs inflammation and embolism, gastrointestinal dysfunction, including diarrhoea, have been reported in 2%–79.1% of Covid-19 cases [9]. Involvement of the gastrointestinal tract in Covid-19 disease is also supported by the presence of SARS-CoV-2 mRNA in stool specimens of patients and differences in the expression of genes activated during inflammation in the gut epithelium in response to SARS-CoV-2 infection in animal models [10].

According to recent epidemiological data, in spite of the similar prevalence of Covid-19 in males and females, males are more susceptible than females to Covid-19 [11–13]. In particular, the WHO (18.1.2021) reports that male patients represent 50.9% and females 49.1% of 18.137 million confirmed Covid-19 cases. At the same time, the ratio of male vs. female mortality is 1.51, as 60.2% of deaths include men, and 39.8% are in female patients. So far, there is no single explanation for this disparity; most probably, several reasons are behind this statistic, observed in all countries and continents included in the analysis with the exception of Pakistan and Australia [12, 14]. Not one but a combination of several reasons is considered to be a possible explanation for sex-dependent Covid-19 susceptibility. Among them, lifestyle habits, time until medical care initiation, the prevalence of comorbidities, smoking habits, sex-dependent differences in immune system responsiveness and expression of genes related to Covid-19 progression very likely contribute to a less effective response to Covid-19 challenge in males compared to females [12, 14, 15].

In addition to the abovementioned gender-associated factors, there is growing evidence implicating the involvement of a protective effect of estrogens in females that is missing in males [16–18]. 17β-estradiol (E2), the most potent estrogen, exerts its effect predominantly nuclear α (ERα) and β (ERβ) estrogen receptors (encoded by the ESR1 and ESR2 genes, respectively) [19] and membrane-bound G protein-coupled receptor-1 (GPER1, also known as GPR30) that via activation of adenylate cyclase, increases intracellular cAMP production and interacts with several signal pathways [20, 21]. While ESR1 is expressed mainly in the breast, ovary and endometrium, ESR2 and GPER1 are widely distributed within the organism, and their expression is observed in many tissues, including the lungs and colon [22, 23].

The protective role of estradiol with respect to Covid-19 can be executed via modulation of the immune system response [13]. Similarly, it has been shown that E2 significantly decreased the SARS-CoV-2 load after E2 treatment in the VERO E6 monkey kidney cell line [24]. E2 treatment has also been reported effective in regulating ACE2 mRNA expression [14, 17]. Part of the rationale for why increased E2 levels can exert a beneficial effect after SARS-CoV-2 infection issues from studies employing probands subjected to estradiol hormone therapy (EHT). It was shown that in women above 50 years of age, EHT reduced the risk of fatality from Covid-19 by more than 50% [25]. Similarly, short-term oral administration of E2

decreased the time to viral clearance in Covid-19 patients [26]. Although ACE2 levels were not monitored in these studies, it has been demonstrated previously that E2 treatment increased ACE2 activity in the plasma [27]. Similarly, upregulation of ACE2 mRNA expression in response to E2 treatment was observed in the human heart [28]. On the other hand, E2 inhibited ACE2 mRNA expression in differentiated normal human bronchial epithelial cells [29], no changes in ACE2 mRNA expression were observed after E2 administration in the VERO E6 monkey kidney cell line [24] and increases in ACE2 expression in response to ERα-mediated signalling were reported in human endothelial cells [30]. Therefore, data obtained under *in vivo* conditions seems to be more consistent than *in vitro* studies, where the results are probably more dependent on cell type.

Most human physiological functions [31] and pathological processes [32], including SARS-CoV-2 internalisation [3], are influenced by the circadian system. The circadian system generates endogenous rhythms with a period close to 24 h and synchronizes them with the external light/dark (LD) cycle. Organisation of the circadian system is hierarchical, with the central oscillator localised in the suprachiasmatic nuclei of the hypothalamus [33] and peripheral oscillators localised in all other tissues of the human body [31]. The regulatory effect of the circadian system is, to some extent, manifested in all human tissues, including those of the pulmonary [34], gastrointestinal [35] and immune [36, 37] systems.

Functioning of the circadian oscillator is based on coordinated expression of clock genes in both central and peripheral oscillators. Clock genes *period* (*per*) and *cryptochrome* (*cry*) possess capacity to negatively influence their transcription after some time lag, which is generated by posttranscriptional modification of *per* and *cry* protein products by casein kinase 1 (CK1). When CK1 activity is saturated, PER and CRY proteins form a PER:CRY complex, which is translocated into the nucleus. In the nucleus PER:CRY heterodimers interfere with the binding of transcriptional factors that promote *per* and *cry* gene expression in the regulatory region E-box. The most studied transcriptional factors involved in this process are brain and muscle arnt-like protein-1 (BMAL1) and circadian locomotor output cycles kaput (CLOCK). Additional loops contribute to the functioning of the basic feedback loop created by clock genes *per* and *cry* and transcription factors BMAL1 and CLOCK. Among them nuclear receptor REV-ERBα strongly contributes to the rhythmic pattern in BMAL1 and CLOCK expression [38]. Except for clock genes, regulatory regions that mediate the influence of the circadian system on the transcriptome are widely spread out in the genome, which is a reason why 81.7% of primate genes exerts a rhythmic pattern [39].

Our previous research demonstrated that several RAS components show daily rhythms with a low amplitude [4]. In particular, we observed a daily rhythm in ACE2 expression in the heart [40] and the ACE/ACE2 mRNA ratio in the rat aorta [41]. To our knowledge, there is no information about the effect of E2 on ACE2 mRNA expression during the whole 24-h cycle in the context of other molecules influencing ACE2-mediated SARS-CoV-2 entry into the cell. Therefore, the aim of the present study was to reveal if E2 administration can influence the 24-h pattern of ACE2 mRNA expression in the lungs and colon of male rats. The expression of ACE2 mRNA was investigated in concert with the expression of ADAM17, TMPRSS2, clock genes and E2 receptors mRNA.

## Methods

Male Wistar rats (21 weeks old, n = 50) were housed under conditions of a light/dark (LD) cycle with lights on at 7 a.m. and access to standard laboratory food and water *ad libitum*. After acclimatisation they were randomly allocated to a control group (n = 25) and an estradiol-influenced group (E2, n = 25). Estradiol was administered in drinking water in a

concentration of 40 ug/kg/day for 7 days. This way of treatment was chosen because previously it has been shown that peroral E2 administration does not cause supraphysiological concentrations at the beginning of treatment [42] and influences mainly night-time E2 levels in the circulation [43]. Animals from the control group received the vehicle used for E2 administration, which was 0.0324% ethanol dissolved in drinking water. During the experiment, the animals' water consumption, anxiety-like behaviour and locomotor activity were monitored as described previously [40].

Tissue sampling began on day 7 after the initiation of E2 treatment during a whole 24-h cycle (S1 Fig), with the first time point at Zeitgeber time 10. The Zeitgeber time is a relativised time reflecting the LD regimen in the animal facility, where ZT0 is defined as the beginning of the light phase, and ZT12 corresponds to the beginning of the dark phase of the LD cycle. Samples were taken at 4-h intervals at ZT10, ZT14, ZT18, ZT22, ZT2 and ZT6. At each time point, 4–5 controls and the same number of E2-treated rats were used. Rats were anaesthetized with isoflurane and subsequently decapitated. A low-intensity red light was used for sample collection during the dark period. Tissue samples were immediately frozen in liquid nitrogen and stored at −80˚C until RNA extraction.

The experimental protocol was approved by the Ethical Committee for the Care and Use of Laboratory Animals at the Comenius University in Bratislava and the State Veterinary Authority of Slovak Republic.

E2 concentration in rat plasma samples and drinking water was measured in duplicates by commercially available enzyme-linked immunosorbent assay kits: Estradiol Elisa (AR E-8800R and FR E-2000, respectively, LND, Germany) according to manufacturer instructions with intra assay precision 3% and measured sensitivity 0.5 pg/ml.

## Cell culture and cultivation

Human cell line DLD1 (ATCC, USA) was cultured in RPMI 1640 medium without phenol red (Gibco, USA) supplemented with FBS without E2 (Biosera, France), penicillin (50 U/ml), streptomycin (50 μl/ml) (Gibco, USA) and ampicillin (50 μg/ml) (Oasis-lab, SR). E2 stock solutions with decreasing concentrations (R1, R2 and R3) were prepared according to the protocol posted at http://dx.doi.org/10.17504/protocols.io.6qpvr6722vmk/v1. After seeding cells into 24 well plate, E2 was administered into each well to reach final concentration 0 nM (vehiculum), 0.1 nM, 1 nM or 10 nM. Cells were maintained at 37˚C in a humidified incubator (Heraeus, USA) containing 5% $CO_2$ for 48 hours before samples were taken.

## Gene expression

To measure mRNA expression, mRNA was extracted from whole lungs or colon tissue samples (70 mg) using RNAzol according to the manufacturer's instructions (MRC, USA).

Isolation of DLD1 cells was performed with the use of RNAzol according to the modified protocol http://dx.doi.org/10.17504/protocols.io.kxygxz12ov8j/v1 of RNAzol manufacturer (MRC, USA).

Synthesis of cDNA from 1 μg of mRNA was carried out using ImProm-II Reverse Transcription System II kit (Promega, USA) and random hexamers according to the manufacturer's instructions.

The miScript SYBRⓇ Green PCR Kit (Qiagen, Germany) was used to measure gene expression by real-time PCR using thermocycler StepOne™ Plus Real-Time PCR System (Applied Biosystems, USA) according to the manufacturer's instructions. Real-time PCR conditions were hot start at 95˚C for 15 min followed by 40 cycles of 94˚C for 15s, 49–55˚C for 30s and 72˚C for 30s. The sequences of the primers used to measure ACE2, ADAM17, TMPRSS2,

BMAL1, PER2, ESR1, ESR2, GPER1, U6 and beta actin mRNA expression are provided in S1 Table. Expression of U6 and beta actin mRNA was used for normalisation.

Estimation of gene expression intensity between tissues was based on a comparison of PCR cycles where the threshold, set at 1 for all genes, crossed the exponential curve as described previously [44].

### Elevated plus-maze test

The elevated plus-maze was made from wood and contained two open ($50 \times 10$ cm) and two enclosed ($50 \times 40 \times 10$ cm) arms placed 50 cm above the floor level. Animals (control males: n = 10, E2-treated males: n = 10) were placed in the central zone (10 x 10 cm) facing the closed arm. Testing started at 11:00 a.m., and animals were recorded for 5 minutes by a Logitech Webcam C930E. Their movement was then analysed with EthoVision XT 16 (Noldus, Netherland) software. The experimental box was cleaned with water after each trial.

The number of entries and time spent in the open arms as well as the ratio of open to total arm entries and time spent (open/total x 100) were used as measures of the animals' state of anxiety. When the centre point of an animal entered the arm or central zone, this was considered as arm or zone entry.

### Statistics

Distribution of data was tested by D'Agostino & Pearson normality test. When the data set did not show normal distribution non-parametric statistic was used.

Plasma levels of E2 between experimental and control group were compared by Mann-Whitey test.

Differences in consolidation of activity expressed as amount of activity during the L (ZT 0–12) phase relative to the D (ZT 12–24) phase, and differences in the cycle threshold of E2 receptor expression between tissues were compared by an unpaired t-test. Differences in water consumption were calculated by paired t-test.

The number of entries and time spent in the open arms of the elevated plus-maze test were analysed using an unpaired t-test (or by Mann-Whitey test for data with a non-parametric distribution).

Daily profile in gene expression, E2 levels and locomotor activity measured during a 24-h cycle were analysed by Cosinor analysis. Data were fitted to a cosine curve with a 24-h period. The goodness of fit (R-value–correlation coefficient) of the approximated curve was determined by analysis of variance (ANOVA). When the fitted cosine curve significantly matched the experimental data, parameters of fit were calculated with 95% confidence limits [40]. To ensure that the daily pattern of gene expression was not imposed by normalisation of gene expression, a Cosinor curve was also fitted to the 24-h pattern of U6 and beta actin expression and subtracted from the U6 and beta actin daily profile.

Gene expression measured in DLD1 cells was analysed by one-way ANOVA (factor: concentration of E2 supplemented to the cells). Differences among groups were analysed by *post hoc* Tukey multiple comparisons test. Dose-dependent effect of E2 administration on ACE2 and TMPRSS2 mRNA expression was analysed by regression analysis.

Differences were considered significant at $P < 0.05$. Data in graphs are presented as average ± standard error of the mean (SEM).

### Results

Intake of E2 in concentration 40 μg/kg/day issued in averaged intake of 20 μg of E2 per rat/day in the experimental group (S2 Fig). Plasma levels of E2 increased accordingly in E2 treated rats

(S3A Fig) at all time points of 24-h cycle (S2 Table). Daily rhythm in E2 plasma levels observed in control rats (S3B Fig) was not disturbed by E2 treatment. Similarly, like in control, we observed a distinct daily rhythm in E2 treated rats with maximum during the dark phase of 24-h cycle (S3C Fig) with increased mesor and amplitude compared to control (S2 Table).

E2 provided in drinking water caused a significant decrease in water consumption in control compared to E2 treated rats (44.7 ± 1.7 ml *vs*. 34.7 ± 1.0 ml, respectively, P < 0.05, paired t-test).

The daily pattern in locomotor activity of rats under synchronized conditions exhibited a clear-cut rhythm in control and E2-treated animals. There was a steep increase in activity during the dark period and a low level of activity during the light phase of the LD cycle, with no significant difference in the mesor, amplitude and acrophase between groups during the whole experiment (Cosinor, S4A Fig). Similarly, we did not observe a significant difference in consolidation of activity expressed as the ratio of activity measured during the L phase and the amount of activity recorded during the D phase of the LD cycle (t-test, S4B Fig), although there was a trend toward improved consolidation in the E2- treated group.

E2-treated rats showed a more anxious profile in the elevated plus-maze test and spent significantly less time in open arms when compared to the control group (P < 0.05, Mann-Whitey test, S5A Fig). The ratio of open to total arm time spent expressed in % (open/total × 100) was also decreased in the E2-treated group compared to the control (P < 0.05, Mann-Whitey test, S5B Fig). The number of open arm entries and the ratio of open to total arm entries (open/total × 100) did not differ significantly (S5C and S55D Fig).

The expression of ACE2 mRNA in the lungs of control rats did not show a rhythmic pattern, however, we observed a rhythmic profile in ACE2 mRNA expression in the colon (Cosinor, S2 Table). E2 administration induced a daily rhythm in ACE2 mRNA expression in the lungs, with a peak at the transition from the dark to the light phase of the LD cycle (Fig 1A), which was not observed in the colon (Fig 1D).

In the lungs, there was a mild trend toward increased ACE2 mRNA expression at the end of the dark phase of the LD cycle in E2-treated rats compared to the control (t-test, P = 0.145). ACE2 mRNA expression in the colon was significantly increased in rats exposed to E2 treatment compared to the control at ZT10. An increase in ACE2 mRNA levels was observed when all samples taken during 24-h were compared (t-test, P < 0.05, S2 Table).

Expression of ADAM17 showed a significant daily rhythm in control and E2-treated rats in both tissues, and E2 administration did not significantly influence this pattern (Fig 1B and 1E, Cosinor, S2 Table).

The expression of TMPRSS2 mRNA did not show a rhythmic pattern in the lungs but we observed a significant daily rhythm in TMPRSS2 mRNA expression in the colon of control rats (Cosinor, S2 Table). E2 induced TMPRSS2 mRNA expression at the transition from the dark to the light phase of the LD cycle in both tissues (t-test, P < 0.05, Fig 1C and 1F). Because of this increase, which was more pronounced in the lungs than in the colon, a daily rhythm with a maximum at the transition from the dark to the light phase of the LD cycle emerged in the lungs (Cosinor, S2 Table). The expression of TMPRSS2 seems to be more responsive to E2 treatment in the lungs than in the colon.

The effect of E2 administration was investigated also under *in vitro* conditions. In accordance with results obtained in animal experiment, we detected a significant influence of E2 administration on ACE2 and TMPRSS2 mRNA expression, while expression of ADAM17 mRNA was not responsive to E2 treatment in DLD1 cells (S6A–S6C Fig). Interestingly, while effect of E2 on ACE2 mRNA showed a positive correlation with E2 concentration, an opposite pattern was observed in expression of TMPRSS2 mRNA. The expression of TMPRSS2 mRNA was much more responsive to low levels compared to high levels of E2 (S7 Fig).

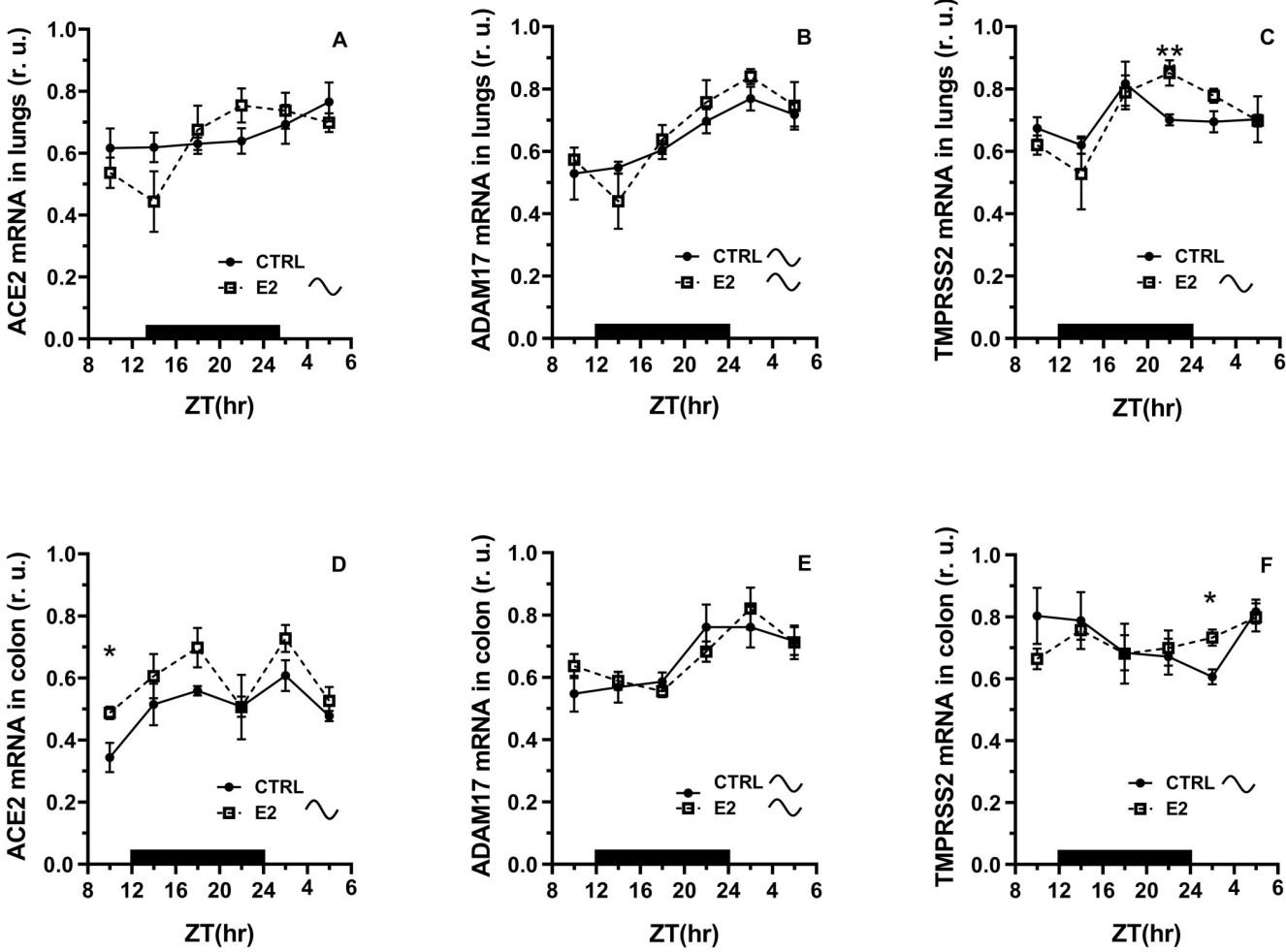

**Fig 1. Effect of 17β-estradiol (E2) administration on the daily pattern of ACE2, ADAM17 and TMPRSS2 mRNA expression in the lungs and colon.**
Rats were synchronized to a 12:12-h LD cycle. The dark bar on the x-axis represents the dark phase of the LD regimen. Full line with circles is attributed to control group, broken line with squares shows data acquired in E2 treated rats. CTRL–control, cosinusoid sign indicates significant daily rhythm in expression (Cosinor), r. u.–relative units. * P < 0.05, ** P < 0.01, unpaired t-test, comparison between time points.

Expression of ESR1 mRNA exerted a significant daily pattern in the control group, which was diminished by E2 administration in the lungs and colon of rats (Fig 2A and 2D, Cosinor, S2 Table). We observed a pronounced trend to increase in ESR1 mRNA expression at the beginning of light phase in the lungs of E2 treated rats compared to control (t-test, Fig 2A, S2 Table). E2 significantly induced ESR1 mRNA expression in the light phase of the LD regimen in the colon (t-test, Fig 2D, S2 Table).

Unlikely ESR1, ESR2 mRNA did not show a significant rhythmic pattern in the examined tissues in control group (Fig 2B and 2E). However, we observed a significant daily rhythm in ESR2 mRNA expression in the lungs of E2 treated rats (Cosinor, S2 Table) which was accompanied by an increase in ESR2 mRNA expression at the beginning of the light phase of the LD cycle (t-test, P < 0.05, Fig 2B). In the colon, we observed a pronounced trend to rhythmic pattern in ESR2 mRNA expression in E2 treated rats (Cosinor, P = 0.07, S2 Table).

Expression of GPER1 mRNA in the lungs and colon did not show a rhythmic pattern in the control groups (Fig 2C and 2F). However, E2 administration induced a rhythm in the daily pattern of GPER1 mRNA levels in the lungs (Fig 2C, Cosinor, S2 Table). A pronounced trend

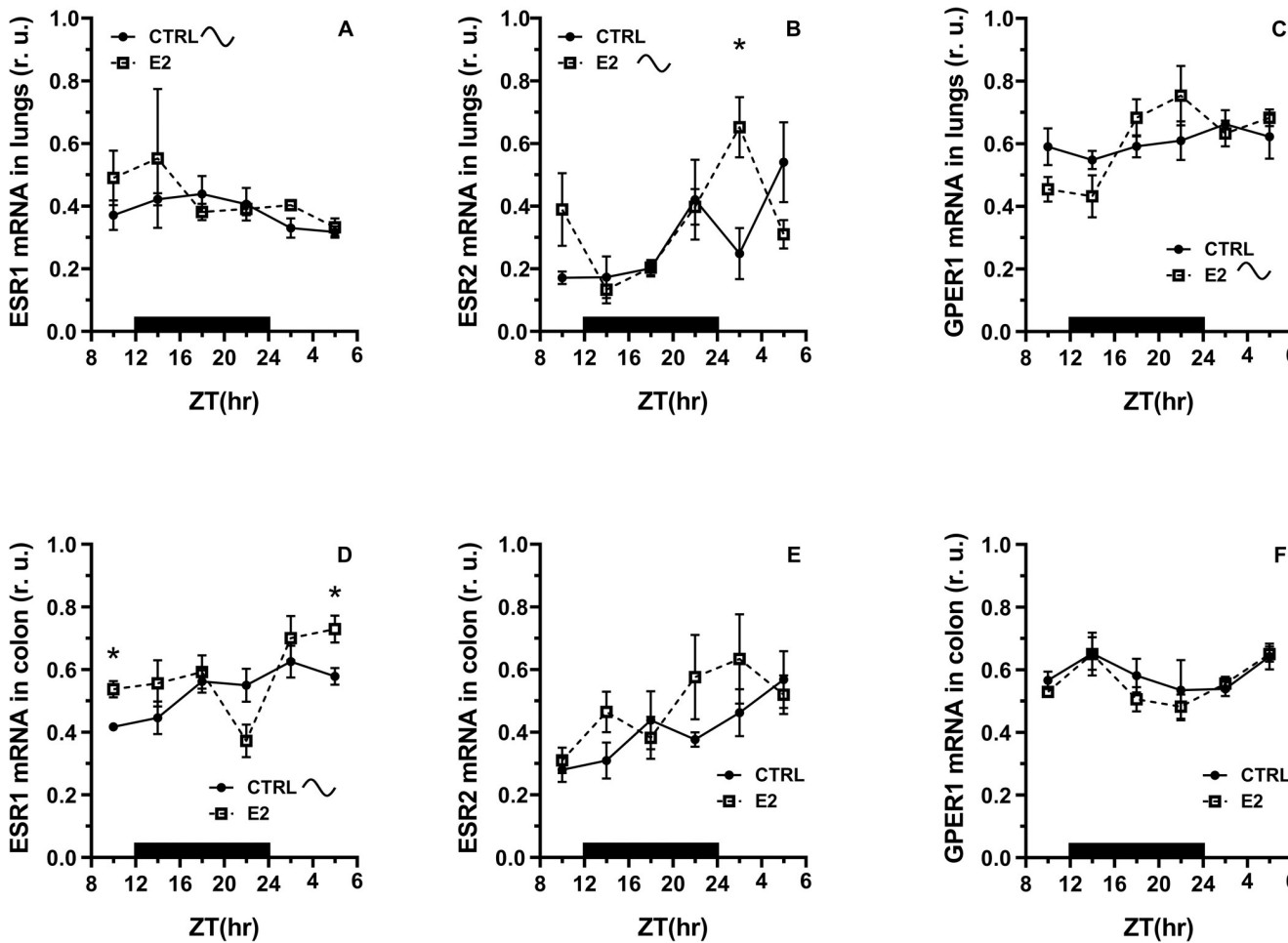

**Fig 2. Effect of 17β-estradiol (E2) administration on the daily pattern in ESR1, ESR2 and GPER1 mRNA expression in the lungs and colon.** Rats were synchronized to a 12:12-h LD cycle. The dark bar on the x-axis represents the dark phase of the LD regimen. Full line with circles is attributed to control group, broken line with squares shows data acquired in E2 treated rats. CTRL–control, cosinusoid sign indicates significant daily rhythm in expression (Cosinor), r. u.–relative units. * P < 0.05, unpaired t-test, comparison between time points.

to rhythmic pattern in ESR2 mRNA expression in E2 treated rats was observed in the colon (Fig 2F, Cosinor, P = 0.08, S2 Table).

Expression of E2 receptors analysed in DLD1 cell culture confirmed responsiveness of ESR2 and GPER1 mRNA expression responsiveness to E2 treatment (S6D–S6F Fig). Expression of ESR1 mRNA was very low in DLD1 cells that caused increase in variability of data. In spite of that, we observed a pronounced trend to increased levels after 10nM E2 treatment compared to control (t-test, P = 0.10).

Cosinor analysis revealed a highly significant rhythmic pattern in PER2 mRNA expression in the lungs and colon that reached maximal levels at the beginning of dark phase of the LD cycle in the control (Cosinor, S2 Table). E2 administration did not significantly influence the rhythm of PER2 mRNA expression in the lungs, however, it caused a significant phase advance in the rhythmic pattern of PER2 mRNA in the colon (Fig 3A and 3C, Cosinor, S2 Table).

Expression of BMAL1 mRNA also showed a clear-cut rhythm with peak levels observed at the transition from the dark to the light phase of the LD cycle (Cosinor, S2 Table). Unlikely the expression of PER2, expression of BMAL1 mRNA was responsive to E2 treatment in both tissues. In the lungs, E2 administration caused an increase in amplitude (Fig 3B, Cosinor,

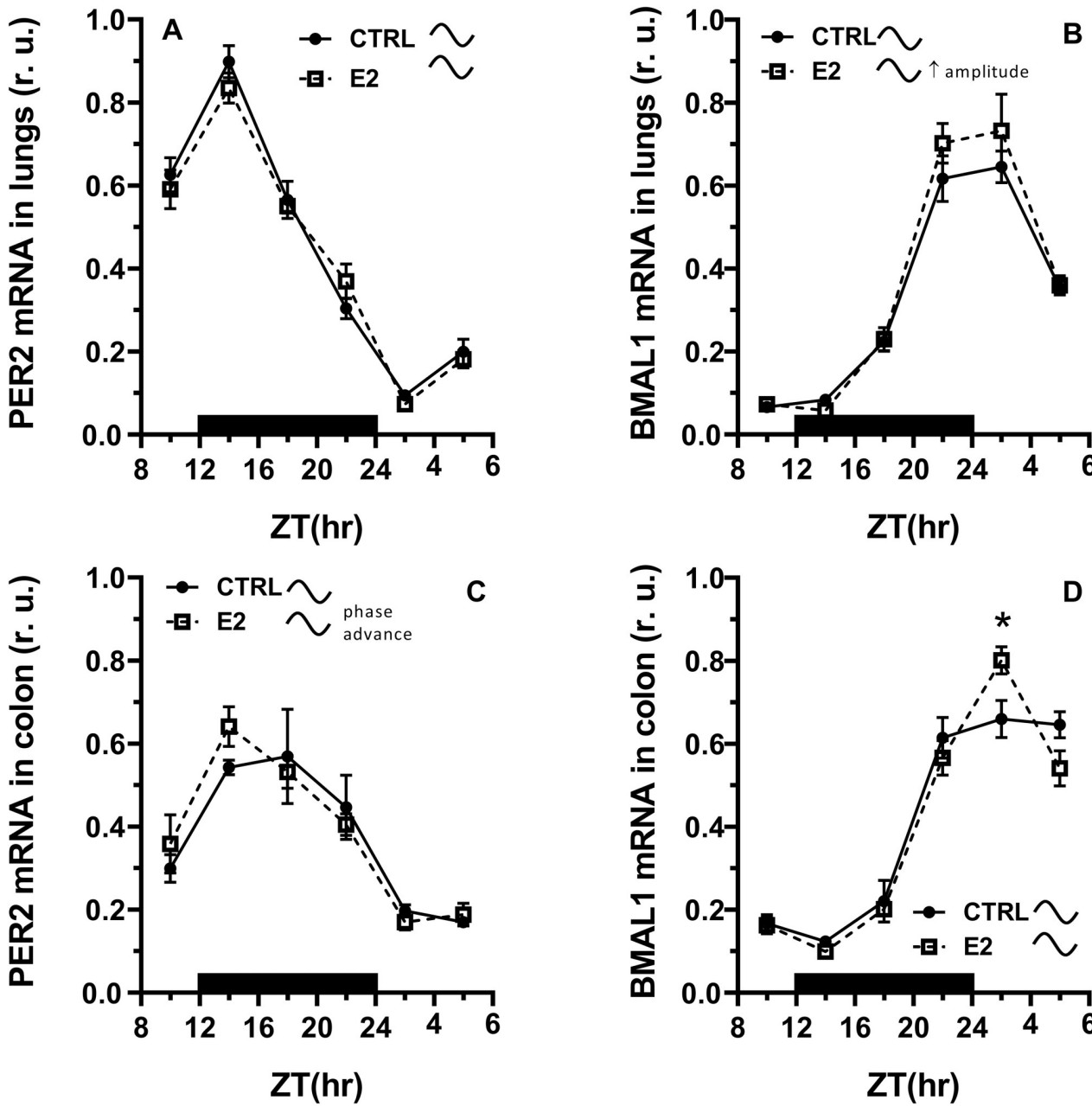

**Fig 3. Effect of 17β-estradiol (E2) administration on the daily pattern in PER2 and BMAL1 mRNA expression in the lungs and colon.** Rats were synchronized to a 12:12-h LD cycle. The dark bar on the x-axis represents the dark phase of the LD regimen. Full line with circles is attributed to control group, broken line with squares shows data acquired in E2 treated rats. CTRL–control, cosinusoid sign indicates significant daily rhythm in expression (Cosinor), r. u.–relative units. * P < 0.05, unpaired t-test, comparison between time points.

S2 Table), and in the colon, we observed an increase in BMAL1 mRNA expression at the very beginning of the light phase of the LD cycle (t-test, P < 0.05, Fig 3D).

In DLD1 cells we observed a pronounced trend to increase in expression of PER2 mRNA after E2 treatment (P = 0.055, S6G Fig). E2 administration strongly influenced BMAL1 mRNA expression in DLD1 cell line (S6H Fig). These results are in line with animal experiment where higher responsiveness of BMAL mRNA expression to E2 treatment compared to PER2 mRNA was observed.

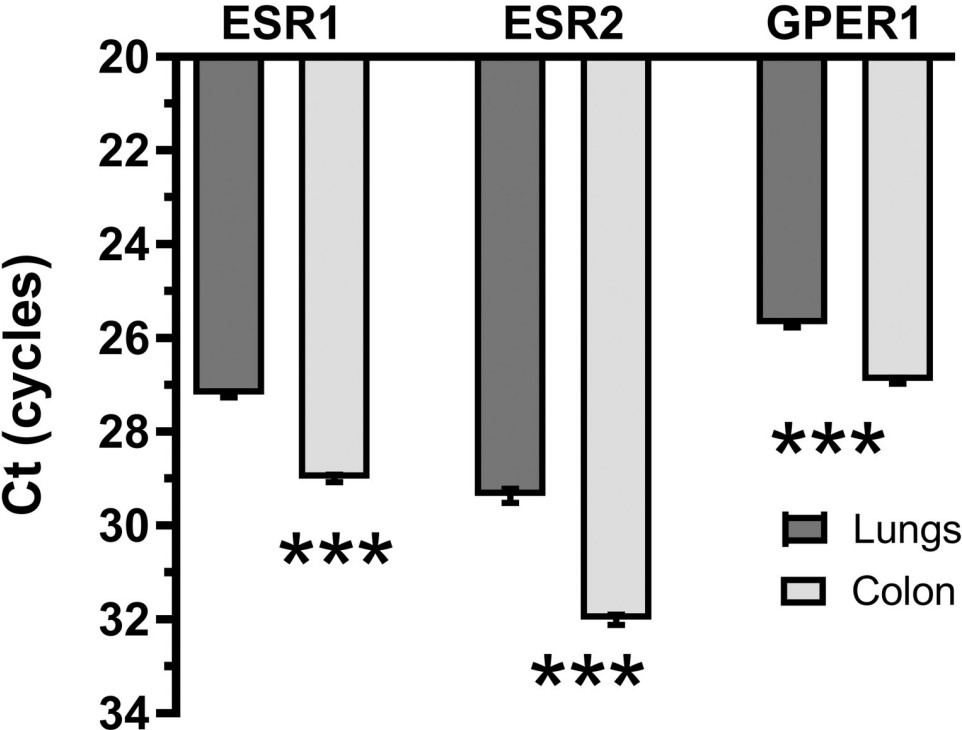

**Fig 4. Relative expression of ESR1, ESR2 and GPER1 mRNA in the lungs and colon.** Relative expression is presented as threshold cycle (Ct) values deducted from the total number of PCR cycles. To reveal differences in Ct between expression of E2 receptors, all control samples measured during 24h cycle were compared between tissues. Ct —the PCR cycle number at which the fluorescence reaches the threshold of amplification. *** P < 0.001, unpaired t-test.

In addition to the daily pattern, we observed significant differences in the intensity of gene expression between tissues as measured at the PCR cycle level where the curve reflecting expression crossed the threshold set at 1 for all genes. The expression of ESR1, ESR2 and GPER1 mRNA was higher in the lungs than in the colon (t-test, P < 0.001, Fig 4).

## Discussion

The present study examined the effect of E2 administration on the daily pattern of ACE2, ADAM17 and TMPRSS2 mRNA expression in the lungs and colon of rats synchronized to a 12:12-h LD cycle. ACE2 and TMPRSS2 mRNA expression was responsive to E2 treatment, which induced a rhythm in the expression of these genes in the lungs. On the other hand, ADAM17 showed a rhythmic profile in the lungs and colon which was not affected by E2 administration. Tissue-specific changes in the expression of members of the SARS-Cov-2 entrance pathway can be related to differences in E2 nuclear and membrane receptors between the lungs and colon. In the lungs, in addition to ACE2 and TMPRSS2, ESR2 and GPER1 mRNA also showed a significant rhythm in their expression after E2 treatment. Increased amplitude in the rhythmic profile of BMAL1 expression after E2 treatment implicates involvement of a peripheral circadian oscillator in the emergence of E2-induced rhythmicity.

The expression of ACE2 mRNA did not show a rhythmic pattern in the lungs, however, we observed a significant daily rhythm in the colon of control animals. A daily pattern in ACE2 mRNA expression in rat lungs has not been investigated previously; however, our data are in accordance with datasets available for mice [45, 46], baboons [39] and humans [45] as well as

the observation that ACE2 mRNA expression measured in a non-small-cell lung cancer cell line under *in vitro* conditions did not show a rhythmic pattern [3].

Previously, a significant daily rhythm in ACE2 mRNA expression in the rat heart and aorta was observed [40, 41]. In the present study, a daily pattern of ACE2 mRNA exhibited a daily rhythm in the colon. It seems that under some circumstances, ACE2 mRNA expression can exert tissue- and/or species-specific rhythmic pattern; however, the amplitude of this rhythm is always very low or shows borderline significance.

Unlikely ACE2, ADAM17 mRNA expression exhibited a clear-cut daily rhythm in the lungs and colon, reaching a maximum during the first half of the light phase of the LD cycle in both tissues. According to available datasets [45], ADAM17 mRNA exerts a daily rhythm in the lungs, kidney and liver in mice and in the visceral fat in humans. Maximal ADAM17 mRNA expression in the lungs was observed at the transition from the dark to the light phase, which is in accordance with the acrophase observed in our study.

TMPRSS2 mRNA expression did not show a daily pattern in the lungs of control group, however, there was a low amplitude rhythmicity in the colon of control animals. According to screenings performed previously, TMPRSS2 mRNA exhibits a daily rhythm in the pituitary gland and SCN in mice with no rhythmic record found in humans [45]. Therefore, as in the case of ACE2, tissue- and species-specific expression most probably influences the daily pattern in TMPRSS2 mRNA levels.

Under physiological conditions, E2 is predominantly produced in the ovaries in females, whereas in males E2 biosynthesis takes place in several tissues [21]. Plasma E2 levels in females display a circadian rhythm which exhibits a rather low amplitude and shows changes in its pattern dependent on the phase of menstrual cycle [47, 48]. Estradiol levels in the circulation of males also show a daily rhythm of low amplitude [49].

Exposure of rats to exogenously administered E2 induced a rhythm in ACE2 and TMPRSS2 mRNA expression in the lungs while E2 administration caused diminishing of rhythmic pattern in ACE2 and TMPRSS2 expression in the colon. Changes observed in the colon are consequence of significant E2 induced increase in ACE2 and TMPRSS2 mRNA expression during light phase of LD cycle. On the other hand, E2 administration did not influence the daily pattern of ADAM17 mRNA expression in either investigated tissue. Therefore, according to our data, E2 exerts a tissue-specific effect on ACE2 and TMPRSS2 mRNA expression, increasing rather than decreasing their levels.

Previously, under *in vitro* conditions, E2 induced ACE2 mRNA expression in differentiated 3T3-L1 adipocytes [50] and human umbilical vein endothelial cells (HUVEC) in a dose-dependent manner [30]. On the other hand, E2 administration caused downregulation of ACE2 mRNA expression in normal human bronchial epithelial cells (NHBE) [29] and in the VERO E6 monkey kidney cell line, E2 did not exert a significant effect on ACE2 mRNA expression [24].

Ovariectomy decreased ACE2 activity in the mouse heart [17] and rat kidney [51], and E2 administration reversed this effect [51]. The opposite effect was observed later, when gonadectomy increased ACE2 activity in the mouse kidney, and this effect was reversed by E2 administration [52]. Finally, ovariectomy did not influence ACE2 activity in the mouse kidney [50]. No effect of ovariectomy and E2 administration on ACE2 activity was observed in the rat heart [51]. These results, however, are not in line with the study where *in vivo* orchiectomy caused a decrease and ovariectomy, an increase in ACE2 activity in rat hearts [53]. Moreover, ERα-mediated stimulation of ACE2 mRNA expression was observed after E2 treatment of human heart tissue slices under *in vitro* conditions [28]. Another tissue investigated in this respect was adipose tissue, where ovariectomy decreased ACE2 activity in mice [50]. Similarly, according to a database search, E2 induced ACE2 mRNA expression in the mouse thymus [54]. A stimulatory effect of E2 on ACE2 mRNA expression is implicated by the observation that both ACE2 expression and E2 levels decrease in parallel with increasing age [54]. The results

mentioned above are certainly influenced to some extent by differences in experimental design and experimental models. In our study, E2 induced a daily rhythm in ACE2 mRNA expression in the lungs; therefore, during some parts of the LD cycle, it was possible to observe an increase in ACE2 mRNA expression while the opposite trend was observed in antiphase. So, it is possible that differences in sampling time contributed to the observed inconsistencies among the abovementioned studies.

As a consequence of E2 treatment, TMPRSS2 mRNA expression in the lungs increased at the transition from the dark to the light phase of the LD cycle, resulting in the emergence of a significant daily rhythm in the lungs. On the other hand, an increase in TMPRSS2 mRNA expression in the colon at the beginning of the light phase of the LD cycle caused diminishing of rhythmic pattern in TMPRSS2 in this tissue.

Previously, it was shown that E2 administration decreases TMPRSS2 mRNA expression in the VERO E6 monkey kidney cell line [24]. On the other hand, E2 administration did not influence TMPRSS2 mRNA expression in NHBE cells [29]. We suppose that the phase-dependent effect of E2 on TMPRSS2 mRNA expression can, at least to some extent, explain the observed differences.

Rhythmic expression of ADAM17 mRNA showed a very similar pattern in the lungs and colon, and its daily profile was not influenced by E2 treatment. This is in accordance with the previous finding that E2 administration to normotensive animals does not cause a change in ADAM17 mRNA expression in the heart [55]. Under *in vitro* conditions, however, the expression of ADAM17 mRNA was upregulated by E2 administration in HUVEC cells pre-treated with interleukin-6 (IL-6) [56]. Similarly, E2 induced ADAM17 mRNA expression in a human non-small-cell lung cancer cell line (NSCLC) in a dose-dependent manner [57]. It should be mentioned that in both *in vitro* studies, the effective concentrations of E2 were much higher compared to that used in animal studies. We did not observe E2 induced ADAM17 mRNA expression in DLD1 cell line.

It is obvious that E2-mediated regulation of members of the SARS-Cov-2 entrance apparatus shows tissue specificity. We suppose that this tissue-specific expression can be, at least to some extent, explained by tissue-specific E2 receptor expression and signalling.

Estradiol nuclear receptors are known to exert direct genomic effects mediated by estrogen response element (ERE), which is widespread throughout the human genome [21]. E2 can regulate ACE2 mRNA expression via two ERE elements that were demonstrated in ACE2 promoter [58]. Moreover, several clusters of multiple EREs have been identified in the ACE2 gene [54]. Similarly, several ERα-binding sites were identified by chromatin interaction analysis close to the TMPRSS2 gene promoter [59].

In addition to genomic regulation [60], the membrane receptor GPER1 can also influence gene expression via several G-protein-coupled downstream pathways [21–23]. Therefore, non-genomic as well as genomic effects of E2 can participate in the changes observed in ACE2 and TMPRSS2 mRNA expression.

Contributing to the complexity of E2-mediated signalling, E2 influences the expression of its own receptors [61, 62], and ERβ inhibits the transcriptional activity of ERα on the ERE-binding domain [63].

In the present study, E2 either caused upregulation of E2 receptors and/or influenced the daily profile of their expression, probably via interference with circadian regulation. ESR1 mRNA expression exerted a significant daily pattern in the lungs and colon of the control group, which was diminished by E2 administration. Although a functional regulatory region E-box has not yet been experimentally proved in the sequence of ESR1 coding gene, its presence is implicated [64]. According to our *in silico* search, a daily rhythm in ESR1 mRNA has been observed previously in the liver, kidney and white adipose tissue of mice [45] with

acrophase observed during the dark phase of the LD cycle, similarly like it was found in our study. ESR1 mRNA also exhibited a rhythmic profile in the coronal artery and visceral fat in humans [45]. Now a direct effect of E2 was manifested by induction of ESR1 mRNA expression at the beginning of the light phase of the LD regimen observed in the colon. Induction of ESR1 expression in response to E2 stimulation [65] has been reported previously in human macrophages [66]. E2-induced stimulation of ESR1 mRNA can be executed via an ERα binding site anchored at the ESR1 gene promoter [59].

Expression of ESR2 mRNA exerted a trend to rhythmic expression in lungs of the control group and a significant daily rhythm in animals exposed to E2. This finding is in accordance with the previous observation that ESR2 expression exhibits a daily rhythm in mouse lungs [67]. Control of the circadian system over ESR2 mRNA expression is executed via a conserved E-box that was identified in the ESR2 promoter. A functional relationship between the circadian system and regulation of ESR2 mRNA expression has also been confirmed by the finding that the rhythm in ESR mRNA expression was abolished in BMAL1 knockout mice [67]. According to the datasets, ESR2 exerts a rhythmic profile in the lungs of mice, with acrophase at the end of the light phase of the LD cycle, similarly to in our study, but there is no such record in humans [45]. We did not observe a rhythmic pattern in ESR2 mRNA expression in the colon, although, there was a pronounced trend to rhythmic pattern in ESR2 mRNA expression in E2 treated rats. The effect of E2 on ESR2 mRNA expression in the lungs and colon may be executed via an ERα binding site that was identified before [59].

According to previous screenings, GPER1 does not show a rhythmic expression profile in human and mouse tissues [45]. Our results are in accordance with this finding, as expression of GPER1 mRNA in the lungs and colon did not show a rhythmic pattern in the control groups. On the other hand, E2 administration induced a rhythm in the daily pattern in the lungs and pronounced trend to rhythmic pattern in the colon. This effect may be related to the presence of ERα binding site that was found close to the GPER1 promoter [59].

There is growing evidence that E2 signalling can influence the circadian system back at several levels of regulation [68]. In our study E2 treatment induced a significant increase in the amplitude of rhythmic BMAL1 mRNA expression in the lungs and an increase in BMAL1 mRNA expression in the colon. This regulation may be mediated by an ERα binding site situated close to the BMAL1 gene promoter identified previously [59]. The functionality of this binding site has, however, not yet been validated.

Our findings are in accordance with the observation that E2 induces BMAL1 and PER2 mRNA expression in ERα-positive HME1 breast epithelial cells, and ERα-knockout causes a pronounced decrease in the amplitude of rhythmic BMAL1 and PER2 mRNA expression in this model [69]. Similarly, E2 induced an increase in PER2 mRNA expression mediated by a conserved ERE present in the PER2 gene promoter in the human embryonic kidney cell line 293T and human breast cancer cell line MCF [70]. PER2 and PER1 mRNA expression was also induced by E2 administration in the SCN and liver of ovariectomised rats [71]. These results are in line in our study as we observed a phase advance in PER2 mRNA expression in the colon of E2 treated rats. Expression of the transcription factor CLOCK, which together with BMAL1 regulates gene expression via E-box, is also upregulated by E2 via ERα [72].

Therefore, we suppose that there is substantial evidence implicating the capacity of E2 to influence peripheral circadian oscillators and that observed differences in the results issue mainly from particular details of the experimental models and designs.

In addition to the role of BMAL1 in the synchronization of clock-controlled genes, its expression has also been linked to regulation of inflammation.

It was shown that *bmal1* deletion disrupted glucocorticoid signalling on the CXCL5 promoter and reduced the efficiency of bacterial clearance in lung epithelial club cells [73]. Several

studies reported that *bmal1* deletion contributes significantly to lung damage and inflammation in response to viral infection [34]. Involvement of BMAL1 in the regulation of neutrophil infiltration and course of influenza infection has been demonstrated lately in knockout animal and cell models. Deletion of *bmal1* from the genome causes, in addition to pronounced changes in the rhythmic transcriptome, elevated pulmonary neutrophilia and a deregulation of reaction to inflammatory stimuli [46]. Majority of the *in vivo* and *in vitro* experimental evidence implicates an association of low levels of BMAL1 with enhanced viral virulence [74, 75]. On the other hand, it was reported that silencing or pharmacological inhibition of BMAL1 reduces SARS-CoV-2 cell entry and replication in human lung Calu-3 epithelial cells [3]. This reduction, according to the authors, resulted from lower availability of ACE2, as a decrease in BMAL1 levels also caused a decrease in ACE2 expression. In spite of these observations, it must be recalled that under *in vivo* conditions decreased levels and/or activity of ACE2 is associated with a worse Covid-19 outcome because of the inhibitory role of ACE2 in AngII-mediated vasoconstriction and proinflammatory effects [1, 6, 7].

Experimental evidence implicating the beneficial effects of E2 for Covid-19 patients is still growing [13, 14, 24, 25]. Whether increased expression of BMAL1 after E2 treatment contributes to the protective influence of E2 reported in Covid-19 management needs to be investigated in more detail. However, as the immune system is under the control of the circadian system [73, 76], a proper synchronisation of organ systems could at least contribute to effective timing of anti-inflammatory drugs. In addition, the role of the circadian system in the treatment of viral infection and Covid-19 specifically issue from experimental as well as epidemiological evidence [77–80].

It is well known that the central oscillator is responsive to E2 signalling [71]. As in our experiment E2 administration did not significantly influence daily pattern in locomotor activity, we suppose that peripheral oscillators are more likely than the central oscillator to be involved in the induction of a rhythmic profile in ACE2, TMPRSS2, ESR2 and GPER1 after E2 administration. E2 treatment caused a pronounced decrease in water intake as reported earlier [81, 82]. We also observed an increase in anxiety-like behaviour in adult male rats treated with E2. E2 induced anxiety-like behaviour is mediated via ERα and ERβ and hypothalamic–pituitary–adrenal axis [83]. GPER1 also mediates anxiety-like behaviour by altering the balance between GABA-ergic and glutamatergic signalling within the basolateral amygdala [84]. Possible effect of E2 on anxiety of patients should be taken in consideration when E2 is to be used in Covid-19 treatment.

As E2 administration is being considered as a supportive therapy in Covid-19 treatment [25, 26] it may be of interest that E2 also influences the circadian system, which in turn regulates the expression of hundreds of clock-controlled genes. The finding that ACE2 and TMPRSS2 mRNA exerted a low-amplitude rhythm in animals exposed to E2 brings up the idea that the daily pattern in their expression can be modified under conditions of rhythmic E2 administration. Although this assumption needs to be tested, it is of special interest in the case of TMPRSS2, as Bromhexine (a TMPRSS2 protease blocker) is already being used in Covid-19 treatment [85].

On the other hand, ADAM17 mRNA expression exerted a clear-cut rhythm insensitive to changes in E2 levels in the circulation. In this respect, ADAM17 could be an interesting molecule for pharmacological targeting, as its manipulation would influence males and females similarly. The use of ADAM17 in Covid-19 treatment has been suggested repeatedly for clinical studies [1, 6, 86]. This process is, however, complicated since ADAM17 is ubiquitously distributed among tissues and regulates plenty of substrates and physiological and pathophysiological processes. On one hand, ADAM17 sheds ACE2 receptors and in this way decreases SARS-CoV-2 entry into the cells, but on the other hand, it promotes TNF release and the 'cytokine storm' [87, 88].

In summary, ACE2 and TMPRSS2 mRNA expression in the lungs of male rats is more responsive to E2 treatment than that in the colon. This observation can be related to the higher expression of E2 receptors in the lungs compared to the colon. Unlike that of ACE2 and TMPRSS2, ADAM17 mRNA expression was not responsive to E2 treatment, and it seems to be rigid in this respect. Whether E2-mediated regulation of ACE2 and TMPRSS2 expression is executed via the ERE domain or indirectly remains to be elucidated.

Our results revealed prominent plasticity in E2 signalling with autoregulatory capacity. E2 administration influenced ESR1 mRNA expression similarly in the lungs and the colon of male rats. Although in most cases E2 induced rhythmicity in gene expression in the lungs, in the case of ESR1, upregulation of its expression induced by E2 treatment led to disappearance of the rhythmic profile in ESR1 mRNA expression. On the other hand, a daily rhythm of expression emerged in ESR2 and GPER1 mRNA in lungs of male rats.

E2 also influences peripheral oscillator functioning, as we observed significant changes in BMAL1 mRNA expression in both the lung and the colon of male rats after E2 administration. In the lungs, E2 administration increased the amplitude of rhythmic BMAL1 mRNA expression, in accordance with the emergence of a rhythmic pattern in the daily profile of ACE2, TMPRSS2, ESR2 and GPER1 mRNA expression. Therefore, a role of peripheral oscillators is implicated in this process (Fig 5).

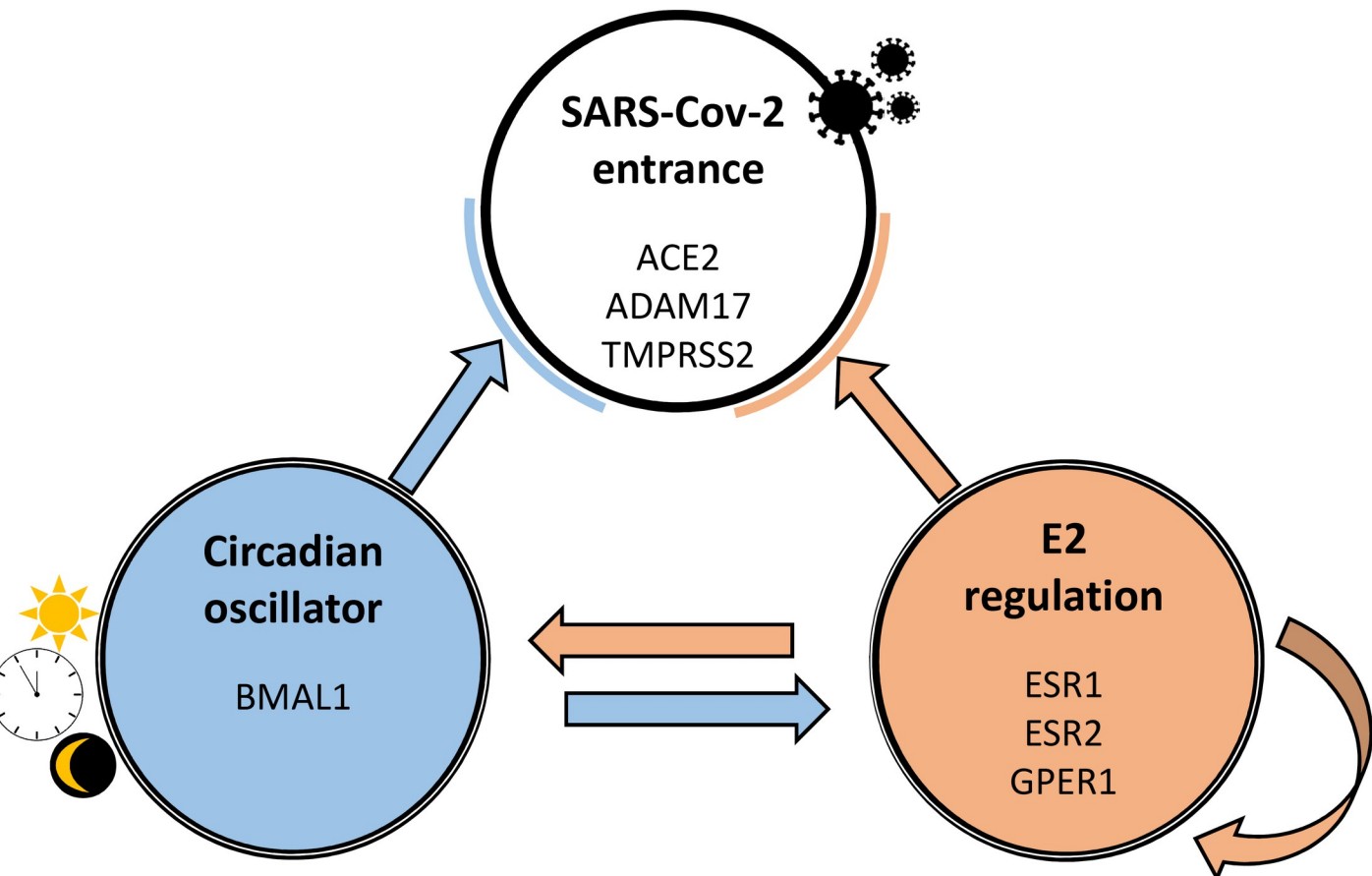

**Fig 5. Regulatory relationship among SARS-Cov-2 entry, the circadian oscillator and E2-mediated signalling supported by experimental evidence.** SARS-Cov-2 entry is influenced both by the circadian system and by E2. E2 signalling is executed via at least three types of E2 receptors and shows autoregulatory capacity. There is also a reciprocal regulatory relationship between the circadian system and E2-mediated regulation. BMAL1 expression seems to be involved in E2-induced changes in peripheral oscillator circadian signalling that can contribute to the emergence of rhythmic profiles in ACE2, TMPRSS2, ESR2 and GPER1. The tissue specificity of this complex regulation can be explained to some extent by higher expression of E2 receptors in the lungs compared to the colon.

Daily rhythms in ACE2, ADAM17 and TMPRSS2 mRNA expression can directly influence SARS-CoV-2 entrance into the cell during the 24-h cycle as ACE2 and TMPRSS2 facilitate this process while ADAM17 decreases levels of membrane-bound ACE2 and in this way inhibits process of SARS-CoV-2 cell entry. Changes in BMAL1 expression have also been associated with efficiency of virulence. Our results implicate that E2 has capacity to influence these processes and therefore its use in Covid-19 treatment should be evaluated with respect to daily pattern in expression of E2-responsive molecules.

## Supporting information

**S1 Fig. Scheme of *in vivo* experiment.** Adult male Wistar rats were synchronized to a light/dark (L, white rectangle; D, black rectangle; respectively) cycle with lights on at 7 a.m. 17β-estradiol (E2) was administered to rats (n = 25) in drinking water in a concentration of 40 μg/kg/day for 7 days. Animals from the control group (n = 25) received the E2 vehicle. Tissue sampling began on day 7 after the initiation of E2 treatment during a whole 24-h cycle, with the first time point at Zeitgeber time 10. ZT0 is defined as the beginning of the light phase, and ZT12 corresponds to the beginning of the dark phase of the LD cycle. Samples were taken at 4-h intervals at ZT10, ZT14, ZT18, ZT22, ZT2 and ZT6. At each time point, 4–5 controls and the same number of E2-treated rats were used.
(TIF)

**S2 Fig. 17β-estradiol (E2) intake of rats.** n = 25, measured E2 concentration in water provided to control is undistinguishable from zero (less than 7.5 pg/rat/day). The x-axis shows time course of experiment.
(TIF)

**S3 Fig. 17β-estradiol (E2) levels in plasma of rats.** Averaged E2 plasma levels in control (white column, n = 25) and E2 treated (grey column, n = 25) rats (A). Daily profile in E2 levels in plasma of control (B, white circles) and E2 treated rats (C, black circles). In some cases SEM is indistinguishable from symbol. Solid line shows significant Cosinor fit (P < 0.05). Black rectangle on x-axis corresponds to dark phase of LD cycle. *** P < 0.001, unpaired t-test.
(TIF)

**S4 Fig. Daily pattern in the locomotor activity of control (solid line) and 17β-estradiol (E2, broken line) treated rats synchronized to a 12:12-h light/dark (LD) cycle.** (A). The black bar at the bottom of the graph represents the dark phase of the LD cycle. (B) L/D is the ratio of activity measured during the light and dark phases of the LD cycle.
(TIF)

**S5 Fig. Estimation of anxiety-like behaviour.** Number of entries and time spent in the open arms of the elevated plus-maze, expressed in absolute and relative units. E2–17β-estradiol, * P < 0.05, Mann-Whitey test.
(TIF)

**S6 Fig.** Effect of 17β-estradiol (E2) administration on ACE2 (A), ADAM17 (B), TMPRSS2 (C), ESR1 (D), ESR2 (E), GPER1 (F), PER2 (G) and BMAL1 (H) mRNA expression in DLD1 cells. NC–negative control. * P < 0.05, ** P < 0.01, ANOVA, post hoc Tukey test.
(TIF)

**S7 Fig. Dose dependent expression of ACE2 (white circles) and TMPRSS2 (gray circles) mRNA in response to 17β-estradiol (E2) administration in DLD1 cells.** Broken line shows significant fit of linear trend in ACE2 and TMPRSS2 mRNA expression. R–regression

coefficient.
(TIF)

**S1 Table. Sequences of the primers used in real-time polymerase chain reaction.** Abbreviations: AT, annealing temperature; ace2, angiotensin converting enzyme 2; adam17, ADAM metallopeptidase domain 17; bAct, beta actin; bmal1, brain and muscle arnt-like protein-1; esr1, oestrogen receptor 1; esr2, oestrogen receptor 2; gper1, G protein-coupled receptor-1; per2, period circadian regulator 2; tmprss2, transmembrane protease/serine protease 2; U6, U6 small nuclear RNA.
(DOC)

**S2 Table. Statistical analysis of gene expression and 17β-estradiol (E2) levels in plasma.** To perform Cosinor analysis a cosine curve with a 24-h period was approximated to the time series data of control and E2-exposed rats. Acrophase–time of curve peak from circadian time zero (dark-to-light transition, ZT0), Mesor–average value of fitted curve, Amplitude–value of curve peak relative to the mesor. Mesor and amplitude are given in relative units. Parameters of rhythms were compared when the fitted curve for both groups showed a significant correlation with the experimental data. P–statistical significance of the fitted cosine curve, R–correlation coefficient, ZT–Zeitgeber time. Gray fields indicate significant differences revealed by Cosinor analysis, * implicates the parameter in which two rhythms differed; trends very close to P < 0.05 are written in italics and a smaller font. ↑ - increase, ↓ - decrease.
(DOC)

## Author Contributions

**Conceptualization:** Iveta Herichová.

**Data curation:** Iveta Herichová.

**Formal analysis:** Iveta Herichová, Soňa Jendrisková, Paulína Pidíková, Lucia Kršková, Lucia Olexová, Martina Morová.

**Funding acquisition:** Iveta Herichová.

**Investigation:** Iveta Herichová, Soňa Jendrisková, Peter Štefánik.

**Methodology:** Iveta Herichová, Lucia Olexová, Martina Morová.

**Project administration:** Iveta Herichová, Lucia Kršková, Lucia Olexová, Katarína Stebelová.

**Supervision:** Iveta Herichová.

**Writing – original draft:** Iveta Herichová.

**Writing – review & editing:** Iveta Herichová, Soňa Jendrisková, Paulína Pidíková, Martina Morová, Katarína Stebelová.

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
