## [Decision Letter · Decision Letter 0]

4 Apr 2022

PONE-D-22-06577Effect of 17β-estradiol on the daily pattern of ACE2, ADAM17, TMPRSS2 and estradiol receptor expression in the lungs and colon of ratsPLOS ONE

Dear Dr. Herichova,

Thank you for submitting your manuscript to PLOS ONE. After careful consideration, we feel that it has merit but does not fully meet PLOS ONE’s publication criteria as it currently stands. Therefore, we invite you to submit a revised version of the manuscript that addresses the points raised during the review process.

Discuss relevance of transcription as readout.

Please provide the rationale of the determination of animal anxiety.

Add transcription and male rats to the title.

Provide the experimental data to confirm the intake amount of E2 in the rats.

Provide the experimental data to validate the activation of ESR downstream genes

Consider shortening introduction and discussion sections.

Provide technical and statistical details as requested by the reviewers.

We look forward to receiving your revised manuscript.

Kind regards,

Henrik Oster, Ph.D.

Academic Editor

PLOS ONE

Journal Requirements:

"Supported by grant APVV-16-0209, APVV-20-0241 and VEGA 1/0679/19. This study was also supported by the Operation Program of Integrated Infrastructure for the project, Advancing University Capacity and Competence in Research, Development and Innovation, ITMS2014+: 313021X329, co-financed by the European Regional Development Fund."

"This work was supported by APVV-16-0209, The Slovak Research and Development Agency, https://www.apvv.sk/?lang=en, to IH; APVV-20-0241, The Slovak Research and Development Agency, https://www.apvv.sk/?lang=en, to IH; VEGA 1/0679/19, Scientific Grant Agency of the Ministry of Education, Science, Research and Sport of the Slovak Republic, https://www.minedu.sk/vedecka-grantova-agentura-msvvas-sra-sav-vega/, to IH;  Operation Program of Integrated Infrastructure for the project, Advancing University Capacity and Competence in Research, Development and Innovation, ITMS2014+: 313021X329, co-financed by the European Regional Development Fund to LO. The funders had no role in study design, data collection and analysis, decision to publish, or preparation of the manuscript. "

Reviewers' comments:

Reviewer's Responses to Questions

**Comments to the Author**

1. Is the manuscript technically sound, and do the data support the conclusions?

Reviewer #1: Partly

Reviewer #2: Partly

Reviewer #3: Yes

2. Has the statistical analysis been performed appropriately and rigorously? 

Reviewer #1: Yes

Reviewer #2: Yes

Reviewer #3: Yes

3. Have the authors made all data underlying the findings in their manuscript fully available?

Reviewer #1: Yes

Reviewer #2: Yes

Reviewer #3: Yes

4. Is the manuscript presented in an intelligible fashion and written in standard English?

Reviewer #1: Yes

Reviewer #2: Yes

Reviewer #3: No

5. Review Comments to the Author

Reviewer #1: Major comments:

The study was aimed at the determination of oscillations in transcription of genes associated with corona-virus attachment to target cells, and the impact of E2 on this behavior. However, I regret the authors did not evaluate how these changes in transcriptional oscillations affected gross protein expression, either by western blotting or immunostaining. After all, this is likely what matters regarding viral homing to target cells and to the susceptibility to severe COVID disease. The oscillation in gene expression is not dramatically affected by E2, thus the importance of these observations is questionable.

Accordingly, in the title "expression" might be replaced by "transcription".

Methods line 187: what was the average daily intake of E2? Could the nocturnal rise in E2 quoted in Ref 44 reflect the circadian nocturnal active phase of the rat, consuming more water at nighttime? In this respect, could documented circadian fluctuations in gene expression might reflect the hour-to hour change in E2 ingestion (likely increasing during the D period), and not necessarily a true physiologic oscillator phenomenon. A mini-pump for drug delivery could overcome this potential technical drawback.

Minor comments:

You may consider quoting Maksimowski, PLoS ONE, 2020, 15, e0241534, Borges, Life Sci. 2017, 191, 1–8 and Chanana, Am. J Physiol. Endocrinol. Metab. 2020, 319, E562–E567, collectively addressing enhanced expression of angiotensin AT1R with testosterone, but enhanced ACE2, as well as AT2R and MASR with estrogen (both serving as receptors to Ang 1-7). You may also quote Abassi, Front physiol 2020, 11, 574753, who further emphasizes the susceptibility of males to develop AKI as compared with female gender and links it to estrogen-derived enhanced expression of the ACE2-Ang 1-7-MasR axis in the female gender

Methods line 219: Please provide the rationale of the determination of animal anxiety.

Methods lines 231-2. I assume L and D refer to light and dark. Please make it clearer, perhaps also adding in parenthesis the ZT.

Reviewer #2: This manuscript presented a study to investigate the effect 17β-estradiol on the expression pattern of Ace2, Tmprss2, Adam17, the estradiol receptors and the clock gene in the lung and colon tissues of male rats. In general, this manuscript could be better written. A number of pitfalls need to be overcome before consideration of publication.

1. Title: Since male rats were used in this study, “male rats” needs to be specified in the title. Also, “male rats” should also be specified in the conclusions.

2. The authors need to explain why male rats were selected for this study.

3. The introduction section is too long and should be more concise.

4. Since E2 was administrated in the drinking water, the authors should provide the experimental data to confirm the intake amount of E2 in the rats.

5. Refer to Point 4, since the water consumption was decreased upon the start of E2 treatment, would the reduced water consumption affect the intake amount of E2 in the rats as previously planned?

6. Refer to Point 4, the authors need to provide the experimental data to validate the activation of ESR downstream genes so as to confirm the successful induction of the E2 treatment in this system. Without this confirmation, it is unknown whether the results could reflect the original study design.

7. Would the reduced water consumption affect the gene expression in the rats?

8. The treatment timing and the sampling timing are complicated. The authors should better provide a schematic diagram to illustrate the treatment and sampling timings.

9. The daily rhythm was only evaluated on Day 7 (Line 193). The authors should explain whether the rhythm on the other days would be similar to that on Day 7. The authors should better provide the experimental data to explain this concern.

10. U6 was used for normalization (Line 215). U6 is better to be the housekeeping gene for normalization of small RNA. For gene expression, the housekeeping gene should better be Gapdh or Actb.

11. Figure 1D: Statistical significant difference was not presented in the chart, which was different from the statement “ACE2 mRNA expression in the colon was significantly increased in rats exposed to E2 treatment (line 270-273).

12. Figure 3: Statistical significant differences were not marked in the charts.

13. The authors need to explain how the daily rhythm of the gene expression could be related to the SARS-CoV-2 infection and the COVID-19 treatment.

14. Line 387-388: The references for the published experimental evidences need to be cited.

15. The authors should also confirm the results in the female rats as the estradiol hormone therapy was given to the female subjects (Line 138-139).

Reviewer #3: To Academic Editor of PLOS One

The manuscript # PONE-D-22-06577 entitled "Effect of 17β-estradiol on the daily pattern of ACE2, ADAM17, TMPRSS2 and estradiol receptor expression in the lungs and colon of rats" by Herichova I et al.

The authors hypothesized that a better Covid-19 survival rate in females can be attributed to the presence of higher 17β-estradiol (E2) levels in women than in men. Based on the fact that cellular entry of SARS-CoV-2 virus is facilitated by the use of ACE2 and the expression of several RAAS components has been shown to exert a rhythmic pattern; the authors aimed to elucidate possible interference between E2 signaling and the circadian system in the regulation of expression of ACE2 mRNA and functionally related molecules. They found that following E2 administration, a rhythmic pattern in molecules facilitating SARS-CoV-2 entry into the cell, clock genes and E2 receptor. They concluded that the daily pattern of components of the SARS-CoV-2 entrance pathway and their responsiveness to E2 should be considered in the timing of pharmacological therapy for Covid-19.

Analysis of the research:

1. Hypothesis rationale: Comprehensive and is based on their experience and other studies.

2. Study design: The power of the study to identify > 5% change among the study groups. The Protocol and the methods are well designed.

3. Results: Very interesting and coincide with the accumulating knowledge of gender differences in the prognosis of COVID patients.

4. Discussion: the discussion is comprehensive and includes comparisons to the relevant articles.

Comments:

1. The introduction is very long and difficult to follow the rationale behind this interesting study. I recommend to summarize the introduction and focus on the importance of gender differences on COVID severity and prognosis and the effect of E2 on rhythmic pattern of molecules facilitating SARS-CoV-2 entry into the cell, clock genes and E2 receptor.

2. Recently, a manuscript that was published and showed that in human organoids, hrsACE2 can significantly block early stages of SARS-CoV-2 infections (Monteil V, Kwon H, Prado P, Hagelkrüys A, Wimmer RA, Stahl M, Leopoldi A, Garreta E, Hurtado Del Pozo C, Prosper F, Romero JP, Wirnsberger G, Zhang H, Slutsky AS, Conder R, Montserrat N, Mirazimi A, Penninger JM. Inhibition of SARS-CoV-2 Infections in Engineered Human Tissues Using Clinical-Grade Soluble Human ACE2. Cell. 2020 May 14;181(4):905-913).

3. The author should clarify whether they examined whole lung lysate or alternatively, they cultures alveolar epithelial cells.

4. Table S2 should be clarified. Data on E2 values should be noted instead of mentioning "ns".

5. Figure 4. The tissue variation doesn't seem, in my eyes, to be circadian.

6. Discussion is too long and doesn't concentrate mainly on the study findings; but also discuss speculations that may weaken the study. I advise the authors to be concise and concentrate more on the interesting findings of this study.

7. On the conclusion section, I would add a sentence regarding future therapy with E2 on COVID patients.

8. Please, go over the English typos and make corrections. For example, line 36 add "the" between "with" and "use". Same wise on line 40, to add "the" between "of" and "expression". Line 356, to delete "a" that follows "significant". Line 524, to replace "is" with "it". Line 594, to correct "influence" to "influenced".

6. PLOS authors have the option to publish the peer review history of their article (what does this mean?). If published, this will include your full peer review and any attached files.

Reviewer #1: No

Reviewer #2: No

Reviewer #3: No

---

## [Author Response · Author response to Decision Letter 0]

31 May 2022

Enclosed, please, find revised version of our manuscript entitled “Herichová I., Jendrisková S., Pidíková P., Kršková L., Olexová L., Morová M., Stebelová K., Štefánik P.: Effect of 17β-estradiol on the daily pattern of ACE2, ADAM17, TMPRSS2 and estradiol receptor transcription in the lungs and colon of male rats”.

We are very grateful for the opportunity to improve our manuscript (MS) and greatly appreciate all comments and suggestions proposed by the editors and reviewers. All suggestions, including 17β-estradiol (E2) measurement and providing of in vitro evidence, have been accepted and incorporated into MS.

We believe that results demonstrating how E2 and the circadian system shape the daily pattern in expression of molecules facilitating SARS-CoV-2 entry into the cell is very up-today and interesting for wide spectrum of readers. 

Our data indicate that E2 and the circadian system interact in regulation of ACE2 and TMPRSS2 mRNA expression while expression of ADAM17 mRNA seems to be mainly under control of the circadian system and exerts a distinct daily rhythm in expression without regard to E2 treatment. Capacity of E2 to influence gene expression is now also supported by in vitro evidence. 

We also revealed significant tissue specific effect of E2 administration on expression of its own receptors. Similarly, E2 administration influenced expression of clock genes per2 and bmal1. 

We believe that the abovementioned results can be useful in Covid-19 management. Moreover, our data can at least partly elucidate inconsistencies reported earlier on E2 induced effects on expression of molecules facilitating SARS-CoV-2 entry into the cell as daily pattern was not taken into consideration previously.

Concerning funding information, we have to acknowledge APVV-16-0209, APVV-20-0241, VEGA 1/0679/1 and the Operation Program of Integrated Infrastructure for the project, Advancing University Capacity and Competence in Research, Development and Innovation, ITMS2014+: 313021X329, co-financed by the European Regional Development Fund.

We appreciate if text provided below is available with the MS (if MS is accepted of course): “Research was supported by grants APVV-16-0209, APVV-20-0241, VEGA 1/0679/19 and by the Operation Program of Integrated Infrastructure for the project, Advancing University Capacity and Competence in Research, Development and Innovation, ITMS2014+: 313021X329, co-financed by the European Regional Development Fund.”

We hope that you will find revised version of manuscript suitable for publishing in your prominent and reputable journal PLOS ONE.

Yours sincerely, 

authors

Comments from the editorial board

Discuss relevance of transcription as readout.

We investigated regulation of ACE2, TMPRSS2 and ADAM17 levels at the transcriptional level as both, the circadian system as well as E2 exert the most of their effect via regulation of transcription. The circadian feed-back loop influences transcription via E-box and E2 via ERE regulatory region. Moreover, according published literature, it is possible to expect that protein translation will follow levels of mRNA. We provide several references showing positive correlation between mRNA and protein levels of studied genes:

ACE2

Gao, H., Tanchico, D. T., Yallampalli, U., & Yallampalli, C. (2016). A Low-Protein Diet Enhances Angiotensin II Production in the Lung of Pregnant Rats but not Nonpregnant Rats. Journal of pregnancy, 2016, 4293431. 

Xue, T., Wei, N., Xin, Z., & Qingyu, X. (2014). Angiotensin-converting enzyme-2 overexpression attenuates inflammation in rat model of chronic obstructive pulmonary disease. Inhalation toxicology, 26(1), 14–22. 

Khoury, E. E., Knaney, Y., Fokra, A., Kinaneh, S., Azzam, Z., Heyman, S. N., & Abassi, Z. (2021). Pulmonary, cardiac and renal distribution of ACE2, furin, TMPRSS2 and ADAM17 in rats with heart failure: Potential implication for COVID-19 disease. Journal of cellular and molecular medicine, 25(8), 3840–3855. 

Bellamine, A., Pham, T., Jain, J., Wilson, J., Sahin, K., Dallaire, F., Seidah, N. G., Durkee, S., Radošević, K., & Cohen, É. A. (2021). L-Carnitine Tartrate Downregulates the ACE2 Receptor and Limits SARS-CoV-2 Infection. Nutrients, 13(4), 1297. 

Martins, F. L., Tavares, C., Malagrino, P. A., Rentz, T., Benetti, A., Rios, T., Pereira, G., Caramelli, B., Teixeira, S. K., Krieger, J. E., & Girardi, A. (2021). Sex differences in the lung ACE/ACE2 balance in hypertensive rats. Bioscience reports, 41(12), BSR20211201. 

Pedrosa, M. A., Valenzuela, R., Garrido-Gil, P., Labandeira, C. M., Navarro, G., Franco, R., Labandeira-Garcia, J. L., & Rodriguez-Perez, A. I. (2021). Experimental data using candesartan and captopril indicate no double-edged sword effect in COVID-19. Clinical science (London, England : 1979), 135(3), 465–481. 

Wang, K., Xu, Y., Yang, W., & Zhang, Y. (2017). Insufficient hypothalamic angiotensin-converting enzyme 2 is associated with hypertension in SHR rats. Oncotarget, 8(12), 20244–20251. 

Zhang, W., Xu, Y. Z., Liu, B., Wu, R., Yang, Y. Y., Xiao, X. Q., & Zhang, X. (2014). Pioglitazone upregulates angiotensin converting enzyme 2 expression in insulin-sensitive tissues in rats with high-fat diet-induced nonalcoholic steatohepatitis. TheScientificWorldJournal, 2014, 603409. 

ADAM17

Khoury, E. E., Knaney, Y., Fokra, A., Kinaneh, S., Azzam, Z., Heyman, S. N., & Abassi, Z. (2021). Pulmonary, cardiac and renal distribution of ACE2, furin, TMPRSS2 and ADAM17 in rats with heart failure: Potential implication for COVID-19 disease. Journal of cellular and molecular medicine, 25(8), 3840–3855. 

Zheng, D. Y., Zhao, J., Yang, J. M., Wang, M., & Zhang, X. T. (2016). Enhanced ADAM17 expression is associated with cardiac remodeling in rats with acute myocardial infarction. Life sciences, 151, 61–69. 

TMPRSS2

Khoury, E. E., Knaney, Y., Fokra, A., Kinaneh, S., Azzam, Z., Heyman, S. N., & Abassi, Z. (2021). Pulmonary, cardiac and renal distribution of ACE2, furin, TMPRSS2 and ADAM17 in rats with heart failure: Potential implication for COVID-19 disease. Journal of cellular and molecular medicine, 25(8), 3840–3855. 

Bellamine, A., Pham, T., Jain, J., Wilson, J., Sahin, K., Dallaire, F., Seidah, N. G., Durkee, S., Radošević, K., & Cohen, É. A. (2021). L-Carnitine Tartrate Downregulates the ACE2 Receptor and Limits SARS-CoV-2 Infection. Nutrients, 13(4), 1297. 

ESR1

Khaksari, M., Hajializadeh, Z., Mahani, S. E., Soltani, Z., & Asadikaram, G. (2021). Estrogen receptor agonists induce anti edema effects by altering α and β estrogen receptor gene expression. Acta neurobiologiae experimentalis, 81(3), 286–294. 

Soares, T. S., Fernandes, S. A., Lima, M. L., Stumpp, T., Schoorlemmer, G. H., Lazari, M. F., & Porto, C. S. (2013). Experimental varicocoele in rats affects mechanisms that control expression and function of the androgen receptor. Andrology, 1(5), 670–681. 

Fernandes, S. A., Gomes, G. R., Siu, E. R., Damas-Souza, D. M., Bruni-Cardoso, A., Augusto, T. M., Lazari, M. F., Carvalho, H. F., & Porto, C. S. (2011). The anti-oestrogen fulvestrant (ICI 182,780) reduces the androgen receptor expression, ERK1/2 phosphorylation and cell proliferation in the rat ventral prostate. International journal of andrology, 34(5 Pt 1), 486–500. 

Harvey, C. N., Chen, J. C., Bagnell, C. A., & Uzumcu, M. (2015). Methoxychlor and its metabolite HPTE inhibit cAMP production and expression of estrogen receptors α and β in the rat granulosa cell in vitro. Reproductive toxicology (Elmsford, N.Y.), 51, 72–78. 

GPER1 (GPR30)

Hsieh, Y. C., Yu, H. P., Frink, M., Suzuki, T., Choudhry, M. A., Schwacha, M. G., & Chaudry, I. H. (2007). G protein-coupled receptor 30-dependent protein kinase A pathway is critical in nongenomic effects of estrogen in attenuating liver injury after trauma-hemorrhage. The American journal of pathology, 170(4), 1210–1218. 

Lo, E., Lee, J. C., Turner, P. C., & El-Nezami, H. (2021). Low dose of zearalenone elevated colon cancer cell growth through G protein-coupled estrogenic receptor. Scientific reports, 11(1), 7403. 

Jala, V. R., Radde, B. N., Haribabu, B., & Klinge, C. M. (2012). Enhanced expression of G-protein coupled estrogen receptor (GPER/GPR30) in lung cancer. BMC cancer, 12, 624. 

PER2

Harbour, V. L., Weigl, Y., Robinson, B., & Amir, S. (2014). Phase differences in expression of circadian clock genes in the central nucleus of the amygdala, dentate gyrus, and suprachiasmatic nucleus in the rat. PloS one, 9(7), e103309. 

Harbour, V. L., Weigl, Y., Robinson, B., & Amir, S. (2013). Comprehensive mapping of regional expression of the clock protein PERIOD2 in rat forebrain across the 24-h day. PloS one, 8(10), e76391. 

Wang, X., Wang, L., Yu, Q., Xu, Y., Zhang, L., Zhao, X., Cao, X., Li, Y., & Li, L. (2016). Alterations in the expression of Per1 and Per2 induced by Aβ31-35 in the suprachiasmatic nucleus, hippocampus, and heart of C57BL/6 mouse. Brain research, 1642, 51–58. 

BMAL1

Asher G, Gatfield D, Stratmann M, Reinke H, Dibner C, Kreppel F, Mostoslavsky R, Alt FW, Schibler U. SIRT1 regulates circadian clock gene expression through PER2 deacetylation. Cell. 2008 Jul 25;134(2):317-28. doi: 

Cha, S., Wang, J., Lee, S.M. et al. Clock-modified mesenchymal stromal cells therapy rescues molecular circadian oscillation and age-related bone loss via miR142-3p/Bmal1/YAP signaling axis. Cell Death Discov. 8, 111 (2022). 

Liu, S., Zhou, Y., Chen, Y., Liu, Y., Peng, S., Cao, Z., & Xia, H. (2022). Bmal1 promotes cementoblast differentiation and cementum mineralization via Wnt/β-catenin signaling. Acta histochemica, 124(3), 151868. 

Please provide the rationale of the determination of animal anxiety.

Nearly all molecules facilitating SARS-CoV-2 entry into the cell have been suggested as therapeutic targets to handle Covid-19 infection. If E2 is also to be used in this way it is of importance to learn which side effects can be expected. It has been previously shown that E2 administration can induce anxiety like behaviour in rodents (Borrow and Handa, 2017), and our results confirmed this earlier observation. This aspect of possible E2 treatment can be relevant and should not be ignored.

Borrow AP, Handa RJ. Estrogen Receptors Modulation of Anxiety-Like Behavior. Vitam Horm. 2017; 103: 27–52.

Add transcription and male rats to the title.

The title of MS has been changed accordingly.

Provide the experimental data to confirm the intake amount of E2 in the rats.

Levels of E2 in drinking water and plasma of experimental and control animals were measured by Elisa. Chapters “Methods” and “Results” were modified accordingly. Supplementary figure 1 showing water consumption was replaced by new figure showing E2 intake. This figure is labelled as supplementary figure 2 now as we incorporated scheme of experiment and sampling, as it was requested by reviewer #2, as a supplementary figure 1.

Provide the experimental data to validate the activation of ESR downstream genes

New in vitro experiment testing dose dependent effect of E2 on ACE2, ADAM17, TMPRSS2, E2 receptors and clock genes was performed as it was requested and incorporated into MS. Chapters “Methods” and “Results” were modified accordingly. Results are summarised in supplementary figures 6 and 7. In vivo and in vitro data are compared in the chapter “Results”. 

Consider shortening introduction and discussion sections.

Chapters “Introduction” and “Discussion” were shortened as it was suggested by reviewers #2 and #3.

Provide technical and statistical details as requested by the reviewers.

All requested statistical and technical details have been incorporated in the chapter “Methods”.

• A rebuttal letter that responds to each point raised by the academic editor and reviewer(s). You should upload this letter as a separate file labelled 'Response to Reviewers'.

• A marked-up copy of your manuscript that highlights changes made to the original version. You should upload this as a separate file labelled 'Revised Manuscript with Track Changes'.

• An unmarked version of your revised paper without tracked changes. You should upload this as a separate file labelled 'Manuscript'.

• We recommend that you deposit your laboratory protocols in protocols.io.

Response to reviewers and manuscript with and without changes were uploaded. We also deposited protocols for isolation of mRNA and miRNA from cells (isolation from single wells of 24 well plate) and protocol for E2 dilution in protocols.io and referred to them in the chapter “Methods”.

All files were named according PLOS ONE's style requirements.

"Supported by grant APVV-16-0209, APVV-20-0241 and VEGA 1/0679/19. This study was also supported by the Operation Program of Integrated Infrastructure for the project, Advancing University Capacity and Competence in Research, Development and Innovation, ITMS2014+: 313021X329, co-financed by the European Regional Development Fund."

"This work was supported by APVV-16-0209, The Slovak Research and Development Agency, https://www.apvv.sk/?lang=en, to IH; APVV-20-0241, The Slovak Research and Development Agency, https://www.apvv.sk/?lang=en, to IH; VEGA 1/0679/19, Scientific Grant Agency of the Ministry of Education, Science, Research and Sport of the Slovak Republic, https://www.minedu.sk/vedecka-grantova-agentura-msvvas-sra-sav-vega/, to IH; Operation Program of Integrated Infrastructure for the project, Advancing University Capacity and Competence in Research, Development and Innovation, ITMS2014+: 313021X329, co-financed by the European Regional Development Fund to LO. The funders had no role in study design, data collection and analysis, decision to publish, or preparation of the manuscript. "

Information about funding has been provided in the cover letter and was deleted from Acknowledgments section.

Reviewer #1:

Major comments:

The study was aimed at the determination of oscillations in transcription of genes associated with corona-virus attachment to target cells, and the impact of E2 on this behavior. However, I regret the authors did not evaluate how these changes in transcriptional oscillations affected gross protein expression, either by western blotting or immunostaining. 

We focused on regulation at the transcriptional level as both, the clock genes as well as E2 exerts most of their effects via regulation of transcription. The circadian feed-back loop is exerting this effect via E-box and E2 via ERE regulatory region. Moreover, according to published literature, it is possible to expect that protein translation will follow levels of mRNA of genes included in the study. We provide several references showing positive correlation between mRNA and protein levels of studied genes:

ACE2

Gao, H., Tanchico, D. T., Yallampalli, U., & Yallampalli, C. (2016). A Low-Protein Diet Enhances Angiotensin II Production in the Lung of Pregnant Rats but not Nonpregnant Rats. Journal of pregnancy, 2016, 4293431. 

Xue, T., Wei, N., Xin, Z., & Qingyu, X. (2014). Angiotensin-converting enzyme-2 overexpression attenuates inflammation in rat model of chronic obstructive pulmonary disease. Inhalation toxicology, 26(1), 14–22. 

Khoury, E. E., Knaney, Y., Fokra, A., Kinaneh, S., Azzam, Z., Heyman, S. N., & Abassi, Z. (2021). Pulmonary, cardiac and renal distribution of ACE2, furin, TMPRSS2 and ADAM17 in rats with heart failure: Potential implication for COVID-19 disease. Journal of cellular and molecular medicine, 25(8), 3840–3855. 

Bellamine, A., Pham, T., Jain, J., Wilson, J., Sahin, K., Dallaire, F., Seidah, N. G., Durkee, S., Radošević, K., & Cohen, É. A. (2021). L-Carnitine Tartrate Downregulates the ACE2 Receptor and Limits SARS-CoV-2 Infection. Nutrients, 13(4), 1297. 

Martins, F. L., Tavares, C., Malagrino, P. A., Rentz, T., Benetti, A., Rios, T., Pereira, G., Caramelli, B., Teixeira, S. K., Krieger, J. E., & Girardi, A. (2021). Sex differences in the lung ACE/ACE2 balance in hypertensive rats. Bioscience reports, 41(12), BSR20211201. 

Pedrosa, M. A., Valenzuela, R., Garrido-Gil, P., Labandeira, C. M., Navarro, G., Franco, R., Labandeira-Garcia, J. L., & Rodriguez-Perez, A. I. (2021). Experimental data using candesartan and captopril indicate no double-edged sword effect in COVID-19. Clinical science (London, England : 1979), 135(3), 465–481. 

Wang, K., Xu, Y., Yang, W., & Zhang, Y. (2017). Insufficient hypothalamic angiotensin-converting enzyme 2 is associated with hypertension in SHR rats. Oncotarget, 8(12), 20244–20251. 

Zhang, W., Xu, Y. Z., Liu, B., Wu, R., Yang, Y. Y., Xiao, X. Q., & Zhang, X. (2014). Pioglitazone upregulates angiotensin converting enzyme 2 expression in insulin-sensitive tissues in rats with high-fat diet-induced nonalcoholic steatohepatitis. TheScientificWorldJournal, 2014, 603409. 

ADAM17

Khoury, E. E., Knaney, Y., Fokra, A., Kinaneh, S., Azzam, Z., Heyman, S. N., & Abassi, Z. (2021). Pulmonary, cardiac and renal distribution of ACE2, furin, TMPRSS2 and ADAM17 in rats with heart failure: Potential implication for COVID-19 disease. Journal of cellular and molecular medicine, 25(8), 3840–3855. 

Zheng, D. Y., Zhao, J., Yang, J. M., Wang, M., & Zhang, X. T. (2016). Enhanced ADAM17 expression is associated with cardiac remodeling in rats with acute myocardial infarction. Life sciences, 151, 61–69. 

TMPRSS2

Khoury, E. E., Knaney, Y., Fokra, A., Kinaneh, S., Azzam, Z., Heyman, S. N., & Abassi, Z. (2021). Pulmonary, cardiac and renal distribution of ACE2, furin, TMPRSS2 and ADAM17 in rats with heart failure: Potential implication for COVID-19 disease. Journal of cellular and molecular medicine, 25(8), 3840–3855. 

Bellamine, A., Pham, T., Jain, J., Wilson, J., Sahin, K., Dallaire, F., Seidah, N. G., Durkee, S., Radošević, K., & Cohen, É. A. (2021). L-Carnitine Tartrate Downregulates the ACE2 Receptor and Limits SARS-CoV-2 Infection. Nutrients, 13(4), 1297. 

ESR1

Khaksari, M., Hajializadeh, Z., Mahani, S. E., Soltani, Z., & Asadikaram, G. (2021). Estrogen receptor agonists induce anti edema effects by altering α and β estrogen receptor gene expression. Acta neurobiologiae experimentalis, 81(3), 286–294. 

Soares, T. S., Fernandes, S. A., Lima, M. L., Stumpp, T., Schoorlemmer, G. H., Lazari, M. F., & Porto, C. S. (2013). Experimental varicocoele in rats affects mechanisms that control expression and function of the androgen receptor. Andrology, 1(5), 670–681. 

Fernandes, S. A., Gomes, G. R., Siu, E. R., Damas-Souza, D. M., Bruni-Cardoso, A., Augusto, T. M., Lazari, M. F., Carvalho, H. F., & Porto, C. S. (2011). The anti-oestrogen fulvestrant (ICI 182,780) reduces the androgen receptor expression, ERK1/2 phosphorylation and cell proliferation in the rat ventral prostate. International journal of andrology, 34(5 Pt 1), 486–500. 

Harvey, C. N., Chen, J. C., Bagnell, C. A., & Uzumcu, M. (2015). Methoxychlor and its metabolite HPTE inhibit cAMP production and expression of estrogen receptors α and β in the rat granulosa cell in vitro. Reproductive toxicology (Elmsford, N.Y.), 51, 72–78. 

GPER1 (GPR30)

Hsieh, Y. C., Yu, H. P., Frink, M., Suzuki, T., Choudhry, M. A., Schwacha, M. G., & Chaudry, I. H. (2007). G protein-coupled receptor 30-dependent protein kinase A pathway is critical in nongenomic effects of estrogen in attenuating liver injury after trauma-hemorrhage. The American journal of pathology, 170(4), 1210–1218. 

Lo, E., Lee, J. C., Turner, P. C., & El-Nezami, H. (2021). Low dose of zearalenone elevated colon cancer cell growth through G protein-coupled estrogenic receptor. Scientific reports, 11(1), 7403. 

Jala, V. R., Radde, B. N., Haribabu, B., & Klinge, C. M. (2012). Enhanced expression of G-protein coupled estrogen receptor (GPER/GPR30) in lung cancer. BMC cancer, 12, 624. 

PER2

Harbour, V. L., Weigl, Y., Robinson, B., & Amir, S. (2014). Phase differences in expression of circadian clock genes in the central nucleus of the amygdala, dentate gyrus, and suprachiasmatic nucleus in the rat. PloS one, 9(7), e103309. 

Harbour, V. L., Weigl, Y., Robinson, B., & Amir, S. (2013). Comprehensive mapping of regional expression of the clock protein PERIOD2 in rat forebrain across the 24-h day. PloS one, 8(10), e76391. 

Wang, X., Wang, L., Yu, Q., Xu, Y., Zhang, L., Zhao, X., Cao, X., Li, Y., & Li, L. (2016). Alterations in the expression of Per1 and Per2 induced by Aβ31-35 in the suprachiasmatic nucleus, hippocampus, and heart of C57BL/6 mouse. Brain research, 1642, 51–58. 

BMAL1

Asher G, Gatfield D, Stratmann M, Reinke H, Dibner C, Kreppel F, Mostoslavsky R, Alt FW, Schibler U. SIRT1 regulates circadian clock gene expression through PER2 deacetylation. Cell. 2008 Jul 25;134(2):317-28. doi: 

Cha, S., Wang, J., Lee, S.M. et al. Clock-modified mesenchymal stromal cells therapy rescues molecular circadian oscillation and age-related bone loss via miR142-3p/Bmal1/YAP signaling axis. Cell Death Discov. 8, 111 (2022). 

Liu, S., Zhou, Y., Chen, Y., Liu, Y., Peng, S., Cao, Z., & Xia, H. (2022). Bmal1 promotes cementoblast differentiation and cementum mineralization via Wnt/β-catenin signaling. Acta histochemica, 124(3), 151868. 

After all, this is likely what matters regarding viral homing to target cells and to the susceptibility to severe COVID disease. 

We did not make any statement about viral homing, however, without mRNA there is no protein to facilitate homing.

The oscillation in gene expression is not dramatically affected by E2, thus the importance of these observations is questionable.

We cannot agree that changes induced by E2 are not dramatic. E2 administration induced significant daily rhythm in ACE2 mRNA and TMPRSS2 mRNA expression in lungs and strongly modulated expression of its own receptors in both investigated tissues. Originality of MS underlines also discovery of daily rhythm in ADAM17 which is present in both, control and experimental group, showing the same acrophase and amplitude that implicates, that this rhythmicity can be expected also in other tissues. All these changes can be too subtle to influence severe Covid disease, however, with omicron variant importance of mild disease rises. Less aggressive variants of SARS-CoV-2 should not be underestimated as they can, similarly like former more aggressive variants, induce long-time Covid symptoms (Morioka et al., 2022).

Post COVID-19 condition of the Omicron variant of SARS-CoV-2; Shinichiro Morioka, Shinya Tsuzuki, Michiyo Suzuki, Mari Terada, Masako Akashi, Yasuyo Osanai, Chika Kuge, Mio Sanada, Keiko Tanaka, Taketomo Maruki, Kozue Takahashi, Sho Saito, Kayoko Hayakawa, Katsuji Teruya, Masayuki Hojo, Norio Ohmagari. medRxiv preprint 2022, DOI: https://doi.org/10.1101/2022.05.12.22274990, 

https://www.medrxiv.org/content/10.1101/2022.05.12.22274990v1

Accordingly, in the title "expression" might be replaced by "transcription".

- “expression” in the title was replaced by “transcription”

Methods line 187: what was the average daily intake of E2? Could the nocturnal rise in E2 quoted in Ref 44 reflect the circadian nocturnal active phase of the rat, consuming more water at nighttime? In this respect, could documented circadian fluctuations in gene expression might reflect the hour-to hour change in E2 ingestion (likely increasing during the D period), and not necessarily a true physiologic oscillator phenomenon. A mini-pump for drug delivery could overcome this potential technical drawback.

As it was reported in earlier studies, in males and females E2 levels in plasma exert a daily rhythm (Bao et al., 2003; Wijetilleka et al., 2016; Rahman et al., 2019) with maximum during the active phase of LD cycle (Bao et al., 2003; Wijetilleka et al., 2016).

In the revised version of MS levels of E2 in drinking water and plasma of experimental and control animals were measured by Elisa as requested. Supplementary figure 1 showing water consumption was replaced by new figure showing E2 intake (S2 Fig). This figure is labelled as supplementary figure 2 now as we incorporated scheme of experiment and sampling as a supplementary figure 1 (it was requested by reviewer #2).

Bao AM, Liu RY, van Someren EJW, Hofman MA, Cao YX, Zhou JN. Diurnal rhythm of free estradiol during the menstrual cycle. Eur J Endocrinol. 2003;148: 227–232.

Rahman SA, Grant LK, Gooley JJ, Rajaratnam SMW, Czeisler CA, Lockley SW. Endogenous Circadian Regulation of Female Reproductive Hormones. J Clin Endocrinol Metab. 2019;104: 6049–6059.

Wijetilleka S, Mon A, Khan M, Joseph F, Robinson A. Circadian Rhythm of Oestradiol: Impact on the Bone Metabolism of Adult Males. In: J Clin Mol Endocrinol. 2016; p. 7.

Similarly, like in humans, we observed a significant daily rhythm in E2 levels in plasma of control rats with the maximum during the active phase of LD cycle. Our aim was not to disturb daily profile in E2 levels in the circulation and to avoid supraphysiological E2 concentrations. We were successful in reaching this goal as E2 treatment increased amplitude and mesor of daily pattern in E2 levels in the circulation, however, acrophase was the same in both groups. Overall E2 levels were approximately 3-4 times higher in E2 treated rats compared to control and we observed a significant increase in E2 levels in every specific time-point during 24h cycle (please, see S2 Table and S3 Fig). 

Mini-pump delivery would very likely diminish naturally occurring daily rhythm in E2 plasma levels.

Minor comments:

You may consider quoting Maksimowski, PLoS ONE, 2020, 15, e0241534, Borges, Life Sci. 2017, 191, 1–8 and Chanana, Am. J Physiol. Endocrinol. Metab. 2020, 319, E562–E567, collectively addressing enhanced expression of angiotensin AT1R with testosterone, but enhanced ACE2, as well as AT2R and MASR with estrogen (both serving as receptors to Ang 1-7). 

You may also quote Abassi, Front physiol 2020, 11, 574753, who further emphasizes the susceptibility of males to develop AKI as compared with female gender and links it to estrogen-derived enhanced expression of the ACE2-Ang 1-7-MasR axis in the female gender

- thank you for the advice, suggested references were incorporated into MS.

Methods line 219: Please provide the rationale of the determination of animal anxiety.

Nearly all molecules facilitating SARS-CoV-2 entry into the cell have been suggested as therapeutic targets to handle Covid-19 infection. If E2 is also to be used in this way it is of importance to learn which side effects can be expected. It has been previously shown that E2 administration can induce anxiety like behaviour in rodents (Borrow and Handa, 2017), and our results confirmed this earlier observation. This aspect of possible E2 treatment can be relevant and should not be ignored. This explanation is now part of the chapter “Discussion” (lines 558 – 559).

Borrow AP, Handa RJ. Estrogen Receptors Modulation of Anxiety-Like Behavior. Vitam Horm. 2017; 103: 27–52.

Methods lines 231-2. I assume L and D refer to light and dark. Please make it clearer, perhaps also adding in parenthesis the ZT.

- thank you for the comment, ZT in parenthesis was added as requested

Reviewer #2:

This manuscript presented a study to investigate the effect 17β-estradiol on the expression pattern of Ace2, Tmprss2, Adam17, the estradiol receptors and the clock gene in the lung and colon tissues of male rats. In general, this manuscript could be better written. A number of pitfalls need to be overcome before consideration of publication.

1. Title: Since male rats were used in this study, “male rats” needs to be specified in the title. Also, “male rats” should also be specified in the conclusions.

- thank you for the comment, using of male rats was specified in the title and in conclusions.

2. The authors need to explain why male rats were selected for this study.

- male rats were used as males have worse Covid-19 prognosis compared to women (Williamson et al., 2020, Brandi et al., 2021, Ma et al., 2021) and according to some authors, it can be related to lower E2 levels (Costeira et al., 2021). Our in vivo experiment mimics situation in human male probands however using of animals allowed us to measure whole 24h cycle in tissue gene expression that is not possible in patients or healthy volunteers. This assumption and supporting references are mentioned in the chapter “Introduction” (94 – 96 and 144 – 149).

Williamson EJ, Walker AJ, Bhaskaran K, Bacon S, Bates C, Morton CE, et al. Factors associated with COVID-19-related death using OpenSAFELY. Nature. 2020;584: 430–436. 

Brandi ML. Are sex hormones promising candidates to explain sex disparities in the COVID-19 pandemic? Rev Endocr Metab Disord. 2021.

Ma Q, Hao ZW, Wang YF. The effect of estrogen in coronavirus disease 2019. Am J Physiol - Lung Cell Mol Physiol. 2021;321.

Costeira R, Lee KA, Murray B, Christiansen C, Castillo-Fernandez J, Lochlainn MN, et al. Estrogen and COVID-19 symptoms: Associations in women from the COVID Symptom Study. PLoS One. 2021;16.

3. The introduction section is too long and should be more concise.

- thank you for the suggestion. The chapter “Introduction” was significantly shortened and we did our best to make it more concise.

4. Since E2 was administrated in the drinking water, the authors should provide the experimental data to confirm the intake amount of E2 in the rats.

Levels of E2 in drinking water and plasma of experimental and control animals were measured by Elisa. Chapters “Methods” and “Results” were modified accordingly. Supplementary figure 1 showing water consumption was replaced by new figure showing E2 intake. This figure is labelled as supplementary figure 2 now as we incorporated scheme of experiment and sampling, as a supplementary figure 1 as it was requested bellow (point 8).

5. Refer to Point 4, since the water consumption was decreased upon the start of E2 treatment, would the reduced water consumption affect the intake amount of E2 in the rats as previously planned?

- as effect of E2 on water intake is known for a long time we expected it and included it into our initial calculations. Therefore, we were able to administer dose very exactly. 

6. Refer to Point 4, the authors need to provide the experimental data to validate the activation of ESR downstream genes so as to confirm the successful induction of the E2 treatment in this system. Without this confirmation, it is unknown whether the results could reflect the original study design.

New in vitro experiment testing dose dependent effect of E2 on ACE2, ADAM17, TMPRSS2, E2 receptors and clock genes was performed as it was requested and incorporated into MS. Chapters “Methods” and “Results” were modified accordingly. Results of in vitro experiment are summarised in the supplementary figures 6 and 7. In vivo and in vitro data are compared in the chapter “Results”. 

7. Would the reduced water consumption affect the gene expression in the rats?

- as water was provided ad libitum and consumed volume of water in E2 treated rats was still perfectly in the physiological range (Claasen et al., 1994) this parameter can hardly influence gene expression measured in this study in the lungs and colon. 

Claassen V.: Techniques in the Behavioral and Neural Sciences, 12 - Food and water intake. Elsevier, Volume 12, 1994, Pages 267-287,

8. The treatment timing and the sampling timing are complicated. The authors should better provide a schematic diagram to illustrate the treatment and sampling timings.

- thank you for the suggestion. Scheme of experiment and sampling was created and referred as supplementary figure 1 mentioned in the chapter “Methods”.

9. The daily rhythm was only evaluated on Day 7 (Line 193). The authors should explain whether the rhythm on the other days would be similar to that on Day 7. The authors should better provide the experimental data to explain this concern.

- the locomotor activity was monitored during the whole experiment. We provide representative data from two days, but the pattern was the same during the whole experiment. As animals were exposed to LD cycle substantial changes in daily pattern of their locomotor activity were not expected and, accordingly, were not observed. This information was provided in the chapter “Results” (line 258). Tissue sampling for gene expression determination requires euthanasia of animals and therefore only one 24h cycle was measured. This is a typical end-point experiment and this design has been well accepted previously, please, see references below:

Herichová I, Tesáková B, Kršková L, Olexová L. Food reward induction of rhythmic clock gene expression in the prefrontal cortex of rats is accompanied by changes in miR-34a-5p expression. Eur J Neurosci. 2021 Nov;54(10):7476-7492. doi: 10.1111/ejn.15518. Epub 2021 Nov 10. PMID: 34735028.

Herichová, I. and Hasáková, K. and Mravec, B. and Kavická, D. Tissue specific regulation of egr1 rhythmic expression in the prefrontal cortex in comparison with other rat brain regions. Activitas Nervosa Superior Rediviva. 2017; 59(2):51-56.

Herichová I, Hasáková K, Lukáčová D, Mravec B, Horváthová Ľ, Kavická D. Prefrontal cortex and dorsomedial hypothalamus mediate food reward-induced effects via npas2 and egr1 expression in rat. Physiol Res. 2017 Dec 30;66(Suppl 4):S501-S510. doi: 10.33549/physiolres.933799. PMID: 29355377.

Zeman M, Molcan L, Herichova I, Okuliarova M. Endocrine and cardiovascular rhythms differentially adapt to chronic phase-delay shifts in rats. Chronobiol Int. 2016;33(9):1148-1160. doi: 10.1080/07420528.2016.1203332. Epub 2016 Jul 26. PMID: 27459109.

Herichová I, Ambrušová J, Molčan Ľ, Veselá A, Svitok P, Zeman M. Different effects of phase advance and delay in rotating light-dark regimens on clock and natriuretic peptide gene expression in the rat heart. Physiol Res. 2014;63(Suppl 4):S573-84. doi: 10.33549/physiolres.932937. PMID: 25669688.

Herichova I, Zsoldosova K, Vesela A, Zeman M. Effect of angiotensin II infusion on rhythmic clock gene expression and local renin-angiotensin system in the aorta of Wistar rats. Endocr Regul. 2014 Jul;48(3):144-51. doi: 10.4149/endo_2014_03_144. PMID: 25110213.

Soltésová D, Veselá A, Mravec B, Herichová I. Daily profile of glut1 and glut4 expression in tissues inside and outside the blood-brain barrier in control and streptozotocin-treated rats. Physiol Res. 2013;62(Suppl 1):S115-24. doi: 10.33549/physiolres.932596. PMID: 24329691.

Herichová I, Šoltésová D, Szántóová K, Mravec B, Neupauerová D, Veselá A, Zeman M. Effect of angiotensin II on rhythmic per2 expression in the suprachiasmatic nucleus and heart and daily rhythm of activity in Wistar rats. Regul Pept. 2013 Sep 10;186:49-56. doi: 10.1016/j.regpep.2013.06.016. Epub 2013 Jul 12. PMID: 23850797.

Šoltésová D, Monošíková J, Koyšová L, Veselá A, Mravec B, Herichová I. Effect of streptozotocin-induced diabetes on clock gene expression in tissues inside and outside the blood-brain barrier in rat. Exp Clin Endocrinol Diabetes. 2013 Aug;121(8):466-74. doi: 10.1055/s-0033-1349123. Epub 2013 Jul 17. PMID: 23864491.

Szántóová K, Zeman M, Veselá A, Herichová I. Effect of phase delay lighting rotation schedule on daily expression of per2, bmal1, rev-erbα, pparα, and pdk4 genes in the heart and liver of Wistar rats. Mol Cell Biochem. 2011 Feb;348(1-2):53-60. doi: 10.1007/s11010-010-0636-x. Epub 2010 Nov 14. PMID: 21076970.

Benova M, Herichova I, Stebelova K, Paulis L, Krajcirovicova K, Simko F, Zeman M. Effect of L-NAME-induced hypertension on melatonin receptors and melatonin levels in the pineal gland and the peripheral organs of rats. Hypertens Res. 2009 Apr;32(4):242-7. doi: 10.1038/hr.2009.12. Epub 2009 Feb 27. PMID: 19262491.

Monosíková J, Zeman M, Veselá A, Herichová I. Down regulation of angiotensin II receptor AT1 expression in the pancreas of diabetic rat. Exp Clin Endocrinol Diabetes. 2009 Sep;117(8):438-9. doi: 10.1055/s-0029-1216376. Epub 2009 May 26. PMID: 19472100.

Monosíková J, Herichová I, Mravec B, Kiss A, Zeman M. Effect of upregulated renin-angiotensin system on per2 and bmal1 gene expression in brain structures involved in blood pressure control in TGR(mREN-2)27 rats. Brain Res. 2007 Nov 14;1180:29-38. doi: 10.1016/j.brainres.2007.08.061. Epub 2007 Sep 6. Erratum in: Brain Res. 2009 Jan 28;1251:296-7. PMID: 17915197.

Zeman M, Szántóová K, Stebelová K, Mravec B, Herichová I. Effect of rhythmic melatonin administration on clock gene expression in the suprachiasmatic nucleus and the heart of hypertensive TGR(mRen2)27 rats. J Hypertens Suppl. 2009 Aug;27(6):S21-6. doi: 10.1097/01.hjh.0000358833.41181.f6. PMID: 19633447.

Zeman M, Szántóová K, Herichová I. Ontogeny of circadian oscillations in the heart and liver in chicken. Comp Biochem Physiol A Mol Integr Physiol. 2009 Sep;154(1):78-83. doi: 10.1016/j.cbpa.2009.05.005. Epub 2009 May 15. PMID: 19447189.

Zeman M, Petrák J, Stebelová K, Nagy G, Krizanova O, Herichová I, Kvetnanský R. Endocrine rhythms and expression of selected genes in the brain, stellate ganglia, and adrenals of hypertensive TGR rats. Ann N Y Acad Sci. 2008 Dec;1148:308-16. doi: 10.1196/annals.1410.069. PMID: 19120123.

Herichová I, Monosíková J, Zeman M. Ontogeny of melatonin, Per2 and E4bp4 light responsiveness in the chicken embryonic pineal gland. Comp Biochem Physiol A Mol Integr Physiol. 2008 Jan;149(1):44-50. doi: 10.1016/j.cbpa.2007.10.006. Epub 2007 Oct 11. PMID: 17996471.

Monosíková J, Herichová I, Mravec B, Kiss A, Zeman M. Effect of upregulated renin-angiotensin system on per2 and bmal1 gene expression in brain structures involved in blood pressure control in TGR(mREN-2)27 rats. Brain Res. 2007 Nov 14;1180:29-38. doi: 10.1016/j.brainres.2007.08.061. Epub 2007 Sep 6. Erratum in: Brain Res. 2009 Jan 28;1251:296-7. PMID: 17915197.

Herichová I, Mravec B, Stebelová K, Krizanová O, Jurkovicová D, Kvetnanský R, Zeman M. Rhythmic clock gene expression in heart, kidney and some brain nuclei involved in blood pressure control in hypertensive TGR(mREN-2)27 rats. Mol Cell Biochem. 2007 Feb;296(1-2):25-34. doi: 10.1007/s11010-006-9294-4. Epub 2006 Aug 15. PMID: 16909304.

Herichová I, Zeman M, Stebelová K, Ravingerová T. Effect of streptozotocin-induced diabetes on daily expression of per2 and dbp in the heart and liver and melatonin rhythm in the pineal gland of Wistar rat. Mol Cell Biochem. 2005 Feb;270(1-2):223-9. doi: 10.1007/s11010-005-5323-y. PMID: 15792371.

Seres J, Herichová I, Roman O, Bornstein S, Jurcovicová J. Evidence for daily rhythms of the expression of proopiomelanocortin, interleukin-1-beta and interleukin-6 in adenopituitaries of male long-evans rats: effect of adjuvant arthritis. Neuroimmunomodulation. 2004;11(5):316-22. doi: 10.1159/000079412. PMID: 15316242.

Roman O, Seres J, Herichova I, Zeman M, Jurcovicova J. Daily profiles of plasma prolactin (PRL), growth hormone (GH), insulin-like growth factor-1 (IGF-1), luteinizing hormone (LH), testosterone, and melatonin, and of pituitary PRL mRNA and GH mRNA in male Long Evans rats in acute phase of adjuvant arthritis. Chronobiol Int. 2003 Sep;20(5):823-36. doi: 10.1081/cbi-120021085. PMID: 14535356.

10. U6 was used for normalization (Line 215). U6 is better to be the housekeeping gene for normalization of small RNA. For gene expression, the housekeeping gene should better be Gapdh or Actb.

We used U6 to normalise gene expression in our previous studies since U6 shows constant expression during 24h cycle.

Herichova I, Reis R, Hasakova K, Vician M, Zeman M. Sex-dependent regulation of estrogen receptor beta in human colorectal cancer tissue and its relationship with clock genes and VEGF-A expression. Physiol Res. 2019 Dec 20;68(Suppl 3):S297-S305.

Hasakova K, Vician M, Reis R, Zeman M, Herichova I. Sex-dependent correlation between survival and expression of genes related to the circadian oscillator in patients with colorectal cancer. Chronobiol Int. 2018 Sep;35(10):1423-1434.

Hasakova K, Vician M, Reis R, Zeman M, Herichova I. The expression of clock genes cry1 and cry2 in human colorectal cancer and tumor adjacent tissues correlates differently dependent on tumor location. Neoplasma. 2018 Nov 15;65(6):986-992.

U6 is transcribed similarly like mRNA and its length allows use random hexamers in transcription. Therefore, methodologically, it can provide internal control for mRNA transcription. However, we appreciate reviewer suggestion and added another housekeeper – beta actin, which was measured in both tissues and cells and used for normalisation along with U6. All graphs have been changed accordingly and statistics was calculated using new data. Using both housekeepers certainly improves quality of results. Information about using two housekeepers was added into chapter „Methods“.

11. Figure 1D: Statistical significant difference was not presented in the chart, which was different from the statement “ACE2 mRNA expression in the colon was significantly increased in rats exposed to E2 treatment (line 270-273).

- thank you for the comment. The difference mentioned in lines 270-273 of former MS version is related to comparison of all control vs. all E2 treated animals and finding, that E2 increased overall ACE2 mRNA expression in colon. As we did not find a decent way how to put this information into the graph (too long text), we changed S2 Table and provided an overview of statistical analysis in additional column. The title of S2 Table has been changed accordingly. 

12. Figure 3: Statistical significant differences were not marked in the charts.

- thank you for the comment, results of Cosinor analysis were added in all graphs as requested. 

13. The authors need to explain how the daily rhythm of the gene expression could be related to the SARS-CoV-2 infection and the COVID-19 treatment.

Thank you for the comment. Text bellow was incorporated into chapter “Discussion” (line 604 – 610): 

Daily rhythms in ACE2, ADAM17 and TMPRSS2 mRNA expression can directly influence SARS-CoV-2 entrance into the cell during the 24-h cycle as ACE2 and TMPRSS2 facilitate this process while ADAM17 decreases levels of membrane-bound ACE2 and in this way inhibits process of SARS-CoV-2 cell entry. Changes in BMAL1 expression have also been associated with efficiency of virulence. Our results implicate that E2 has capacity to influence these processes and therefore its use in Covid-19 treatment should be evaluated with respect to daily pattern in expression of E2-responsive molecules.

Moreover, there is a paragraph in chapter “Discussion” (lines 528 – 538) describing in detail effect of changed BMAL1 mRNA expression on virulence and symptoms related to Covid-19 disease:

“It was shown that bmal1 deletion disrupted glucocorticoid signalling on the CXCL5 promoter and reduced the efficiency of bacterial clearance in lung epithelial club cells [73]. Several studies reported that bmal1 deletion contributes significantly to lung damage and inflammation in response to viral infection [34]. Involvement of BMAL1 in the regulation of neutrophil infiltration and course of influenza infection has been demonstrated lately in knockout animal and cell models. Deletion of bmal1 from the genome causes, in addition to pronounced changes in the rhythmic transcriptome, elevated pulmonary neutrophilia and a deregulation of reaction to inflammatory stimuli [46]. Majority of the in vivo and in vitro experimental evidence implicates an association of low levels of BMAL1 with enhanced viral virulence [74,75]. On the other hand, it was reported that silencing or pharmacological inhibition of BMAL1 reduces SARS-CoV-2 cell entry and replication in human lung Calu-3 epithelial cells [3].”

The role of the circadian system in the Covid-19 management is supported by references 73 - 80 in the chapter “Discussion” (lines 546 – 549):

“…the immune system is under the control of the circadian system [73,76], a proper synchronisation of organ systems could at least contribute to effective timing of anti-inflammatory drugs. In addition, the role of the circadian system in the treatment of viral infection and Covid-19 specifically issue from experimental as well as epidemiological evidence [77–80].”

14. Line 387-388: The references for the published experimental evidences need to be cited.

- thank you for the comment. We apologise for misunderstanding. A sentence from lines 387-388 (former MS) was just supposed to introduce next 4 paragraphs (lines 418 – 458 recent MS) discussing our results and results of references 30 and 50-58. To avoid this misunderstanding sentence “Unfortunately, previously published experimental evidence focused on the effect of E2 on the expression of ACE2, ADAM17 and TMPRSS2 is rather scarce and not completely consistent.“ was deleted from revised version of MS. All references were preserved.

15. The authors should also confirm the results in the female rats as the estradiol hormone therapy was given to the female subjects (Line 138-139).

- sentence from lines 138-139 “Part of the rationale for why increased E2 levels can exert a beneficial effect after SARS-CoV-2 infection issues from studies employing probands subjected to estradiol hormone therapy (EHT). It was shown that in women above 50 years of age, EHT reduced the risk of fatality from Covid-19 by more than 50%.“ referring to the paper of Seeland at al., 2020 was used just to show origins of idea that E2 administration to Covid-19 patients could be beneficial. However, in clinical trials focused on possible E2 benfits for Covid-19 patients, both sexes are being tested (e.g. NCT04865029 and NCT04853069). The major goal of recent MS is to reveal if E2 administration to males can change their internal milliou in such a way that males would be able coop with Covid-19 disease better. Our results implicate that expression of genes coding molecules determing efficientcy of SARS-CoV-2 spread are influenced by E2 and that single point study cannot reveal complexity of this regulation. 

Reviewer #3: 

To Academic Editor of PLOS One

The manuscript # PONE-D-22-06577 entitled "Effect of 17β-estradiol on the daily pattern of ACE2, ADAM17, TMPRSS2 and estradiol receptor expression in the lungs and colon of rats" by Herichova I et al.

The authors hypothesized that a better Covid-19 survival rate in females can be attributed to the presence of higher 17β-estradiol (E2) levels in women than in men. Based on the fact that cellular entry of SARS-CoV-2 virus is facilitated by the use of ACE2 and the expression of several RAAS components has been shown to exert a rhythmic pattern; the authors aimed to elucidate possible interference between E2 signaling and the circadian system in the regulation of expression of ACE2 mRNA and functionally related molecules. They found that following E2 administration, a rhythmic pattern in molecules facilitating SARS-CoV-2 entry into the cell, clock genes and E2 receptor. They concluded that the daily pattern of components of the SARS-CoV-2 entrance pathway and their responsiveness to E2 should be considered in the timing of pharmacological therapy for Covid-19.

Analysis of the research:

1. Hypothesis rationale: Comprehensive and is based on their experience and other studies.

2. Study design: The power of the study to identify > 5% change among the study groups. The Protocol and the methods are well designed.

3. Results: Very interesting and coincide with the accumulating knowledge of gender differences in the prognosis of COVID patients.

4. Discussion: the discussion is comprehensive and includes comparisons to the relevant articles.

Comments:

1. The introduction is very long and difficult to follow the rationale behind this interesting study. I recommend to summarize the introduction and focus on the importance of gender differences on COVID severity and prognosis and the effect of E2 on rhythmic pattern of molecules facilitating SARS-CoV-2 entry into the cell, clock genes and E2 receptor.

- thank you for the suggestion. Chapter “Introduction” was shortened substantially. We focused mainly on the role of E2 in generating gender differences in Covid-19 prognosis. E2 modulated expression of molecules facilitating SARS-CoV-2 entry into the cell, clock genes and E2 receptors were pointed out in this context.

2. Recently, a manuscript that was published and showed that in human organoids, hrsACE2 can significantly block early stages of SARS-CoV-2 infections (Monteil V, Kwon H, Prado P, Hagelkrüys A, Wimmer RA, Stahl M, Leopoldi A, Garreta E, Hurtado Del Pozo C, Prosper F, Romero JP, Wirnsberger G, Zhang H, Slutsky AS, Conder R, Montserrat N, Mirazimi A, Penninger JM. Inhibition of SARS-CoV-2 Infections in Engineered Human Tissues Using Clinical-Grade Soluble Human ACE2. Cell. 2020 May 14;181(4):905-913).

- than you for the suggestion, this interesting article was incorporated in revised version of MS.

3. The author should clarify whether they examined whole lung lysate or alternatively, they cultures alveolar epithelial cells.

- thank you for the comment. Gene expression in tissues was analysed in samples from the whole lungs or colon. Updated MS includes also in vitro study using DLD1 cell line. Chapter “Methods” was modified to avoid any misunderstanding.

4. Table S2 should be clarified. Data on E2 values should be noted instead of mentioning "ns".

- thank you for the comment. All results of statistical analysis have been provided. As Reviewer #2 suggested to use another housekeeping gene, numbers slightly differ from the previous version of MS, however, major results were not influenced by adding new housekeeper.

5. Figure 4. The tissue variation doesn't seem, in my eyes, to be circadian.

- thank you for the comment. Figure 4 actually does not show a circadian rhythm. Instead it shows relative expression of E2 receptors. We changed Figure 4 description to avoid misunderstanding. In case, that reviewer meant graph D from figure 2 we provide graph shown together with Cosinor fit and details of statistical analysis.

Graph shows averaged ESR1 mRNA expression (empty circles connected with dotted line) and SEM together with cosinor fit that significantly corresponds with experimental data 

(P = 0.005, S2 Table).

Best-fit values 

m 0.531

a 0.09164

c -0.1678

Std. Error 

m 0.01718

a 0.02473

c 0.2603

95% CI (asymptotic) 

m 0.4954 to 0.5667

a 0.04035 to 0.1429

c -0.7075 to 0.3719

Goodness of Fit 

Degrees of Freedom 22

R squared 0.3843

Sum of Squares 0.1618

Sy.x 0.08576

Constraints 

m m > 0

a a > 0

c c < 6.28

Number of points 

# of X values 30

# Y values analyzed 25

6. Discussion is too long and doesn't concentrate mainly on the study findings; but also discuss speculations that may weaken the study. I advise the authors to be concise and concentrate more on the interesting findings of this study.

- thank you for very much for suggestion. Chapter “Discussion” was rewritten. We did our best to omit speculations that can be avoided and focused on major findings of the study.

7. On the conclusion section, I would add a sentence regarding future therapy with E2 on COVID patients.

- thank you for the comment, text below was added in the section “Conclusions” (line 604 – 610).

“Daily rhythms in ACE2, ADAM17 and TMPRSS2 mRNA expression can directly influence SARS-CoV-2 entrance into the cell during the 24-h cycle as ACE2 and TMPRSS2 facilitates this process while ADAM17 decreases levels of membrane bound ACE2 and in this way inhibits process of SARS-CoV-2 cell entry. Changes in BMAL1 expression have also been associated with efficiency of virulence (also requested by reviewer #2). Our results implicate that E2 has capacity to influence these processes and therefore its use in Covid-19 treatment should be evaluated with respect to daily pattern in expression of E2-responsive molecules.”

8. Please, go over the English typos and make corrections. For example, line 36 add "the" between "with" and "use". Same wise on line 40, to add "the" between "of" and "expression". Line 356, to delete "a" that follows "significant". Line 524, to replace "is" with "it". Line 594, to correct "influence" to "influenced".

- thank you for the comment, we did our best to correct all typos, whole MS was carefully controlled.

---

## [Decision Letter · Decision Letter 1]

14 Jun 2022

Effect of 17β-estradiol on the daily pattern of ACE2, ADAM17, TMPRSS2 and estradiol receptor transcription in the lungs and colon of male rats

PONE-D-22-06577R1

Dear Dr. Herichova,

We’re pleased to inform you that your manuscript has been judged scientifically suitable for publication and will be formally accepted for publication once it meets all outstanding technical requirements.

Kind regards,

Henrik Oster, Ph.D.

Academic Editor

PLOS ONE

Additional Editor Comments (optional):

Congrats to this nice story.

Reviewers' comments:

Reviewer's Responses to Questions

**Comments to the Author**

1. If the authors have adequately addressed your comments raised in a previous round of review and you feel that this manuscript is now acceptable for publication, you may indicate that here to bypass the “Comments to the Author” section, enter your conflict of interest statement in the “Confidential to Editor” section, and submit your "Accept" recommendation.

Reviewer #2: All comments have been addressed

Reviewer #3: All comments have been addressed

2. Is the manuscript technically sound, and do the data support the conclusions?

Reviewer #2: Partly

Reviewer #3: Yes

3. Has the statistical analysis been performed appropriately and rigorously? 

Reviewer #2: Yes

Reviewer #3: Yes

4. Have the authors made all data underlying the findings in their manuscript fully available?

Reviewer #2: Yes

Reviewer #3: Yes

5. Is the manuscript presented in an intelligible fashion and written in standard English?

Reviewer #2: Yes

Reviewer #3: Yes

6. Review Comments to the Author

Reviewer #2: The authors have satisfactorily addressed most of the previously raised concerns in the revised manuscript.

Reviewer #3: The authors have satisfactorily addressed my concerns; The introduction and Discussion were summarized and focused mainly on the interference between E2 signaling and circadian variation of ACE2 mRNA and its related functional molecules.

The reference was cited.

The authors reported that the experiments were performed in whole cell lysate

Tabl2 S2 was corrected

The authors clarified Figure 4

7. PLOS authors have the option to publish the peer review history of their article (what does this mean?). If published, this will include your full peer review and any attached files.

Reviewer #2: **Yes: **Tsz Kin Ng

Reviewer #3: No

---

## [Editor Report · Acceptance letter]

20 Jun 2022

PONE-D-22-06577R1 

Effect of 17β-estradiol on the daily pattern of ACE2, ADAM17, TMPRSS2 and estradiol receptor transcription in the lungs and colon of male rats 

Dear Dr. Herichova:

I'm pleased to inform you that your manuscript has been deemed suitable for publication in PLOS ONE. Congratulations! Your manuscript is now with our production department. 

Kind regards, 

on behalf of

Prof. Henrik Oster 

Academic Editor

PLOS ONE